# Molecular basis for the binding and selective dephosphorylation of Na$^+$/H$^+$ exchanger 1 by calcineurin

Ruth Hendus-Altenburger [1,5], Xinru Wang [2,3,5], Lise M. Sjøgaard-Frich[4], Elena Pedraz-Cuesta[4], Sarah R. Sheftic[2], Anne H. Bendsøe[1,4], Rebecca Page[2], Birthe B. Kragelund [1], Stine F. Pedersen [4] & Wolfgang Peti [2]

Very little is known about how Ser/Thr protein phosphatases specifically recruit and dephosphorylate substrates. Here, we identify how the Na$^+$/H$^+$-exchanger 1 (NHE1), a key regulator of cellular pH homeostasis, is regulated by the Ser/Thr phosphatase calcineurin (CN). NHE1 activity is increased by phosphorylation of NHE1 residue T779, which is specifically dephosphorylated by CN. While it is known that Ser/Thr protein phosphatases prefer $p$Thr over $p$Ser, we show that this preference is not key to this exquisite CN selectivity. Rather a combination of molecular mechanisms, including recognition motifs, dynamic charge-charge interactions and a substrate interaction pocket lead to selective dephosphorylation of $p$T779. Our data identify T779 as a site regulating NHE1-mediated cellular acid extrusion and provides a molecular understanding of NHE1 substrate selection by CN, specifically, and how phosphatases recruit specific substrates, generally.

---

[1] Structural Biology and NMR Laboratory, Department of Biology, University of Copenhagen, Ole Maaløes Vej 5, DK-2200 Copenhagen N, Denmark. [2] Department of Chemistry and Biochemistry, University of Arizona, 1041 E. Lowell St., Tucson, AZ 85721, USA. [3] Department of Molecular Biology, Cell Biology and Biochemistry, Brown University, 185 Meeting St, Providence, RI 02912, USA. [4] Cell Biology and Physiology, Department of Biology, University of Copenhagen, Universitetsparken 13, DK-2100 Copenhagen Ø, Denmark. [5] These authors contributed equally: Ruth Hendus-Altenburger, Xinru Wang. Correspondence and requests for materials should be addressed to R.P. (email: rebeccapage@email.arizona.edu) or to B.B.K. (email: bbk@bio.ku.dk) or to S.F.P. (email: sfpedersen@bio.ku.dk) or to W.P. (email: wolfgangpeti@email.arizona.edu)

The plasma membrane Na$^+$/H$^+$ exchanger 1 (NHE1, SLC9A1) is a central regulator of cellular pH and volume, and thus is important for cell proliferation, survival, and motility in mammalian tissues. As a consequence, its dysregulation leads to disease, especially cancer and cardiovascular disorders[1–3]. NHE1 comprises a 12 transmembrane-helix domain that mediates Na$^+$/H$^+$ exchange and a C-terminal intracellular domain that is heavily phosphorylated and functions as a protein: protein interaction hub. Critically, these modifications and interactions control NHE1 activity. The C-terminal ~135 residues of NHE1 (NHE1ct; residues I680-Q815) is an intrinsically disordered region (IDR) that contains most of the identified NHE1 phosphorylation sites[4,5].

Ser/Thr kinases use protein:protein interactions and sequence-specific recognition sequences to identify and phosphorylate their substrates; indeed, for this reason >10,000 kinase phosphorylation sites have been identified, allowing the functional importance of these sites to be readily investigated. Multiple mechanisms regulate NHE1 activity including increases in the free, intracellular Ca$^{2+}$ concentration ([Ca$^{2+}$]$_i$)[6,7], and phosphorylation by the MAP kinases ERK1/2, p38, and c-Jun N-terminal kinase (JNK)[8–10]. We recently discovered that ERK2-mediated NHE1 phosphorylation is achieved, in part, via direct binding between ERK2 and the NHE1ct, resulting in ERK2-mediated phosphorylation of six Ser/Thr residues that conform to the ERK2 recognition sequence: [S/T]P[11].

Much less is known about the specificity controlling the reversing dephosphorylation reactions, which is reflected by the fact that only ~400 phosphatase:substrate pairs are currently known. This is due, in part, to an apparent lack of phosphatase-specific recognition sequences, making it impossible to use bioinformatics to understand the substrate:enzyme relationship[12]. NHE1 interacts with, and is likely regulated by, protein phosphatase 1[13] and 2A[14] and the Tyr phosphatase SHP2[15]. A direct interaction of NHE1 with calcineurin (CN, protein phosphatase 2B or PP3) was also demonstrated[16]. However, if NHE1 is a direct substrate of any of these phosphatases, which residues are dephosphorylated and if this regulation has consequences for NHE1 function is still unknown.

CN is a ubiquitously expressed Ca$^{2+}$-dependent Ser/Thr phosphatase[17]. It regulates multiple physiological processes including development, cardiac function, and the immune response[18]. CN is a heterodimer composed of calcineurin A (CNA) and B (CNB), where CNA can be further divided into four functional domains: the catalytic domain, the CNB-binding domain, a calmodulin (CaM)-binding domain, and an auto-inhibitory domain (AID), which blocks the CN active site[19,20]. Upon an increase in [Ca$^{2+}$]$_i$, both CNB and CaM become Ca$^{2+}$-loaded, ultimately displacing the AID from the catalytic site, stimulating phosphatase activity. However, how CN recognizes its substrates is only poorly understood.

Recently, it was shown that CN recruits regulators, inhibitors, and substrates using two short linear motifs (SLiMs), the PxIxIT and the LxVP motifs[12,21]. These SLiMs are typically found in intrinsically disordered proteins/regions (IDPs/IDRs), and bind to the corresponding PxIxIT and LxVP-binding pockets in CN. The PxIxIT motif binds the catalytic domain of CNA[22], whereas the LxVP motif binds to a hydrophobic cleft at the interface of the CNA and CNB subunits[21]. In contrast to the PxIxIT-binding pocket, the binding cleft for the LxVP motif is only accessible in the active conformation of CN[23]. The interaction between NHE1 and CN was shown to depend on a PxIxIT motif in the NHE1ct[16]. However, it is not known if NHE1ct also has an LxVP motif. Further, if and how these motifs influence substrate recognition and specificity, or if there are other features that define substrate specificity is currently

unknown for CN specifically and for Ser/Thr phosphatases generally.

Here, we integrate multiple complementary molecular and cellular experiments to show how CN is recruited to NHE1. Furthermore, we identify an NHE1 phosphorylation site, T779, that is critical for regulating NHE1 transport activity. Further, we show that CN specifically dephosphorylates $p$T779 to return NHE1 activity to basal levels. Finally, using molecular and cellular experiments, we define the key interactions that generate this CN selectivity at the CN active site. Together, these data not only reveal a key role for phosphatases in controlling NHE1 activity, but also provide molecular insights into the multiple mechanisms used by CN to achieve substrate specificity.

## Results

**NHE1 binds CN using a PxIxIT and an LxVP motif.** NHE1 was previously shown to bind CN using a PxIxIT motif, $^{715}$PVITID$^{720}$ [16]. This sequence is present in the intrinsically disordered C-terminal tail of NHE1 (residues I680-Q815, hereafter referred to as NHE1ct; Fig. 1a; variants used in this study are summarized in Supplementary Fig. 1; CN overview is summarized in Fig. 1b)[5]. An alignment of 15 diverse NHE1ct sequences from multiple species revealed the presence of a putative LxVP motif, $^{684}$LTVP$^{687}$ (human) ~30 amino acids N-terminal to the PxIxIT motif (Fig. 1c). Further, the alignment showed that the LxVP motif is considerably more conserved than the PxIxIT motif, supporting its potential importance in CN binding and NHE1 function.

To test if the putative LxVP motif in NHE1 ($^{684}$LTVP$^{687}$) binds directly to CN, we used NMR spectroscopy. An overlay of the 2D [$^1$H,$^{15}$N] HSQC spectrum of $^{15}$N-labeled NHE1ct in the presence and absence of CN showed that multiple peaks disappeared upon complex formation (Fig. 2a). Specifically, cross-peaks corresponding to NHE1 residues $^{684}$LTVP$^{687}$ (L684, T685, V686) and $^{715}$PVITID$^{720}$ (I717, I719) were significantly broadened partially beyond detection upon binding CN. In addition, many of the peaks originating from the 27 residues that connect the NHE1 LxVP and PxIxIT motifs were also broadened upon complex formation (Fig. 2a, b), indicating either that this region is involved in the interaction with CN or that the conformational freedom of the linker is impaired by the anchoring of both SLiMs to CN[24,25]. Thus, NHE1 $^{684}$LTVP$^{687}$ is a CN-specific LxVP SLiM that, together with NHE1 $^{715}$PVITID$^{720}$, binds to CN.

To determine the contribution of each SLiM to CN binding, we used isothermal titration calorimetry (ITC; Table 1). CN and NHE1ct formed a tight 1:1 complex ($K_D = 59 \pm 9$ nM; Fig. 2c). Mutating either the NHE1 LxVP (NHE1ct$_{ATAP}$) or PxIxIT (NHE1ct$_{AVATAA}$) motifs resulted in a significant reduction in binding affinity (Fig. 2d, e). Specifically, NHE1ct$_{ATAP}$ bound CN with a $K_D$ of $4.8 \pm 0.5$ μM (~80-fold reduction vs. WT), while NHE1ct$_{AVATAA}$ bound CN with a $K_D$ of $2.4 \pm 0.5$ μM (~40-fold reduction vs. WT). Similarly, short NHE1 LxVP and PxIxIT peptides bound CN with $K_D$ values of $4.3 \pm 0.5$ and $6.5 \pm 0.5$ μM, respectively (Fig. 2f, g). The increase in the $K_D$ for the NHE1 peptides versus the corresponding binding compromised variant (NHE1ct$_{ATAP}$ includes only a functional PxIxIT site; NHE1ct$_{AVATAA}$ includes only a functional LxVP site) suggests a contribution by the linker and/or distal residues to the overall binding affinity. Finally, the large increase in affinity observed for NHE1ct compared to the affinities of each peptide or NHE1ct motif variants demonstrates avidity.

**NHE1:CN interaction in cells.** To determine if both the LxVP and PxIxIT motifs are important for NHE1:CN binding in cells, we stably expressed either full-length NHE1$_{WT}$, NHE1$_{ATAP}$,

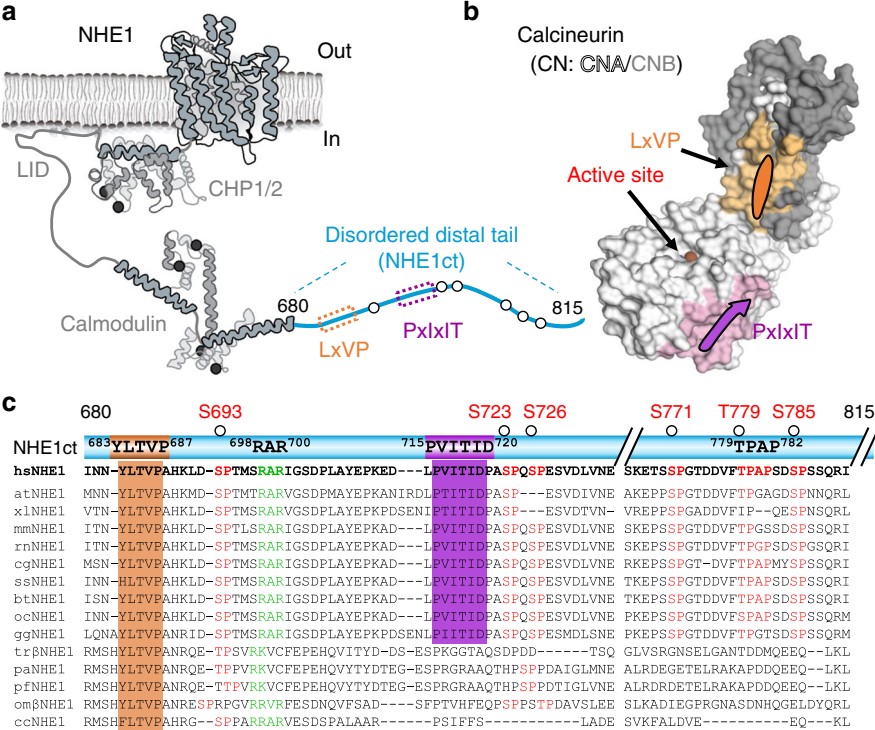

**Fig. 1** Two canonical docking motifs mediate the interaction of NHE1ct with CN. **a** Cartoon illustrating the major structural elements of NHE1, including the C-terminal disordered distal tail (NHE1ct). The locations of the LxVP (dark orange) and PxIxIT (magenta) motifs and ERK2 phosphorylation sites (circles) are highlighted. **b** Cartoon illustrating the major elements of CN, including binding sites for the LxVP (light orange) and PxIxIT (light pink) motifs and the metals (spheres) bound at the active site. **c** Sequence conservation of the LxVP and PxIxIT motifs, the 27-residue linker including the RAR motif (see later) and the ERK2-mediated phosphorylation sites in NHE1ct. Species abbreviations are defined in the "Methods" section

NHE1$_{AVATAA}$, or NHE1$_{AVATAA/ATAP}$ in PS120 mammalian fibroblasts (Fig. 2h, Supplementary Fig. 2; these cells lack endogenous NHE1). As expected from in situ proximity ligation assays (PLA), which confirmed the close proximity (<40 nm) of endogenous NHE1 with endogenous CN in MCF-7 breast cancer cells (Fig. 2i, j), stably expressed WT NHE1 robustly co-immunoprecipitated with CN in PS120 cells, with and without an increase in cellular $[Ca^{2+}]_i$ (Fig. 2h, Supplementary Fig. 2). Similar amounts of NHE1 co-immunoprecipitated with CN for the variants NHE1$_{ATAP}$ and NHE1$_{AVATAA}$, whereas NHE1$_{AVATAA/ATAP}$ co-immunoprecipitated less with CN than WT NHE1. These data together with our NMR and ITC data (Fig. 2a–g) demonstrate that both the NHE1 LxVP and PxIxIT motifs contribute to CN binding.

**Sequence differences in CN-binding motifs fine-tune CN binding.** To understand how NHE1 binds CN at a molecular level, we used ITC and X-ray crystallography (Fig. 3a, b). First, we used ITC to show that NHE1 residues I680-S723 constitute the minimal NHE1 CN-binding domain (NHE1 residues I680-S723, hereafter referred to as NHE1ct$_{\Delta92}$; $K_D = 21 \pm 2$ nM; Fig. 3b; Table 1). Next, we determined the three-dimensional structure of the NHE1ct$_{\Delta92}$:CN triple complex to a resolution of 1.9 Å (CN: CNA$_{1-370}$, CNB$_{16-170}$; Supplementary Table 1). In addition to the electron density observed for CN, strong electron density was observed for NHE1 residues N681-A688 (LxVP motif) and D713-S723 (PxIxIT motif) (Fig. 3a). The NHE1 LxVP and PxIxIT motifs bind CN in the canonical LxVP and PxIxIT-binding pockets, both of which are ~35 Å distal to the CN active site.

The interaction of NHE1 with the CN PxIxIT-binding pocket, which buries 626 Å$^2$ of solvent accessible surface area (SASA), is

similar to that observed in other CN–PxIxIT complexes (Fig. 3c). Namely, the NHE1 PxIxIT motif forms a β-strand that hydrogen bonds with β14 of CNA, extending one of its central β-sheets. Typically, the PxIxIT interaction is further stabilized by multiple hydrophobic contacts between the two conserved I residues. A comparison of the NHE1–PxIxIT interaction with CN to other CN–PxIxIT complexes revealed that the hydrophobic interaction is unique and non-optimal (Fig. 3d). Specifically, in the NHE1-CN complex, both NHE1 PxIxIT residues T718 and D720 form hydrogen bonds with CNA N330. This alters the NHE1 backbone conformation (likely facilitated by the restricted Φ/Ψ space due to NHE1 P721), causing the sidechain of I719 to project out of the hydrophobic CNA PxIxIT-binding pocket (CNA M290, Y288, I331). To confirm this, we mutated the NHE1 PxIxIT motif to a canonical, optimal sequence (PVIVIT), which bound CN five-fold more tightly than WT NHE1ct$_{\Delta92}$ (Supplementary Fig. 3A; Table 1). This non-optimal hydrophobic interaction of NHE1 I719 with CN explains the unusually weak binding of the NHE1 PxIxIT motif to CN, as measured using ITC and co-IP experiments.

The interaction of NHE1 with CN at the LxVP-binding pocket, which buries 560 Å$^2$ of SASA, is essentially identical to those observed between CN and other LxVP motifs[21]. The interaction is dominated by NHE1 LxVP residues L684 and V686, which bind CN in two deep hydrophobic pockets formed at the CNA/B interface (Fig. 3e). Interestingly, and different to the interaction observed between CN and the NFATc1-LxVP$_{peptide}$[26], residues N-terminal to the NHE1 LxVP motif also contribute to binding, albeit in a slightly different manner. Specifically, NHE1 N682 (−2 relative to LxVP) forms a weak hydrogen-bond with CNB Q50, while NHE1 N681 (−3) forms a hydrogen-bond with CNA K360. This shows that residues flanking LxVP motifs also contribute to CN binding.

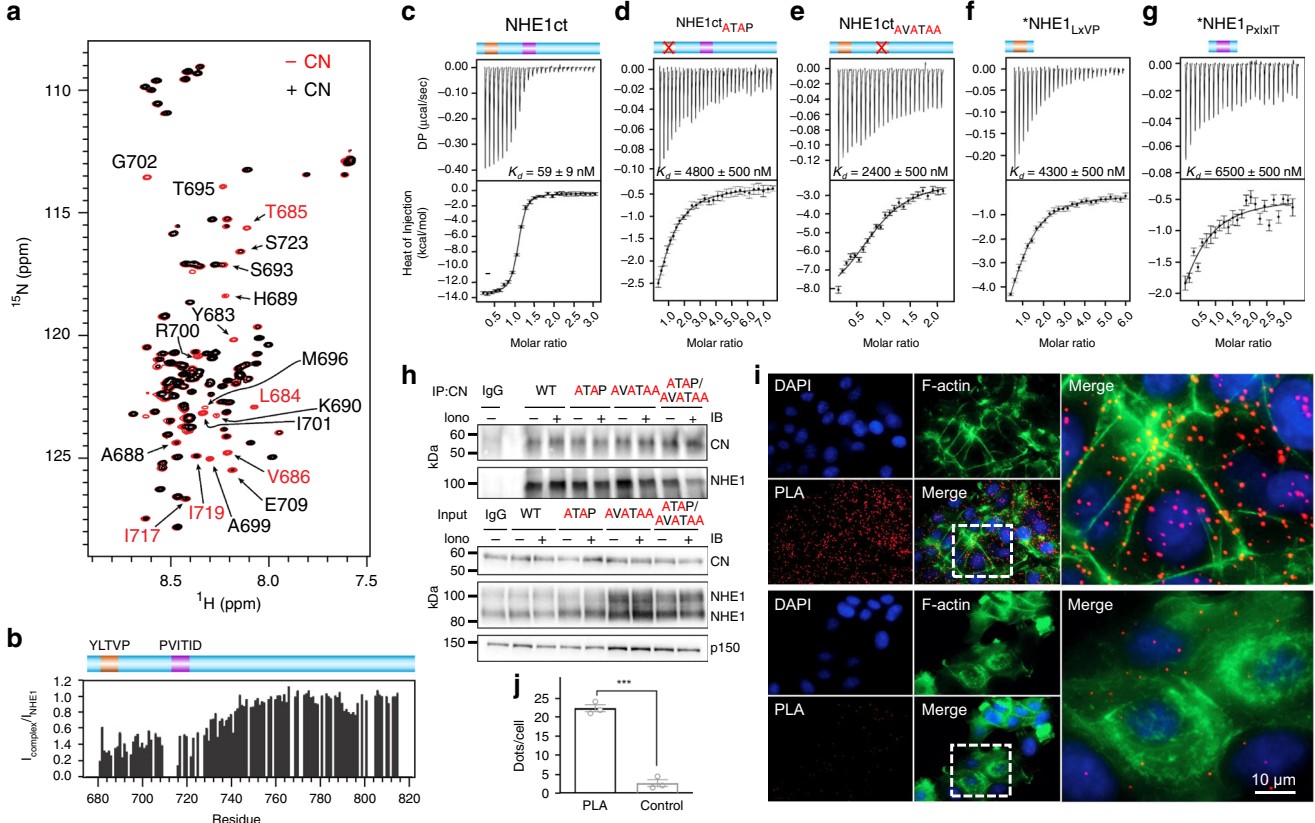

**Fig. 2** NHE1 binding to CN. **a** Overlay of the 2D [$^1$H,$^{15}$N] HSQC spectra of $^{15}$N-labeled NHE1ct in the presence (black) and absence (red) of CN (1:1 ratio). Peaks corresponding to residues from the docking motifs or linker region are labeled (red and black, respectively). **b** [$^1$H,$^{15}$N] HSQC peak intensity ratios for spectra shown in **a**. ITC data for CN with **c** NHE1ct WT ($n = 3$), **d** NHE1ct$_{ATAP}$ ($n = 2$), **e** NHE1ct$_{AVATAA}$ ($n = 2$), **f** *LxVP peptide (NHE1 Q678-P694) ($n = 2$), and **g** *PxIxIT peptide (NHE1 E709-P727) ($n = 2$); * for panels **f** and **g** indicates the use of a peptide. **h** PS120 fibroblasts stably expressing NHE1 and variants, with or without a 10 min pre-treatment with ionomycin to increase intracellular [Ca$^{2+}$]$_i$, were subjected to immunoprecipitation (IP) using anti-CNA antibody ($n = 4$). IP and input were blotted for NHE1 and CN. P150 is shown as loading control. Quantification of data from **h** is shown in Supplementary Fig. 2. **i** Proximity ligation assay (PLA) of NHE1 and CN in MCF7 breast cancer cells ($n = 3$). PLA signal is seen in red, and cells were stained for nucleus (DAPI) and F-actin. PLA was carried out using anti-NHE1 (NHE1–54) and anti-CNA (top panel) and only anti-NHE1 as a control (bottom panel). **j** Quantification of the results in **i** using ImageJ. ***Indicate a *p*-value of <0.001 compared to the control, unpaired Student's *t*-test. Error bars represent S.E.M. values. Source data are provided as a Source Data file

**Table 1 Isothermal titration calorimetry (ITC) measurements of NHE1ct with CN**

| NHE1ct[a] | $K_D$ (nM) | $\Delta H$ (kcal mol$^{-1}$) | $T\Delta S$ (kcal mol$^{-1}$) |
|---|---|---|---|
| *NHE1ct (NHE1 I680-Q815)* | | | |
| NHE1ct WT[b] | 59 ± 9 | −12 ± 2 | −2.5 ± 1.4 |
| *NHE1ct motif variants* | | | |
| NHE1ct$_{AVATAA}$ | 2400 ± 500 | −7.6 ± 0.7 | 0.9 ± 0.2 |
| NHE1ct$_{ATAP}$ | 4800 ± 500 | −4.5 ± 0.7 | −2.8 ± 0.8 |
| *NHE1 motif peptides* | | | |
| PxIxIT (NHE1 E709-P727) | 6500 ± 500 | −2.9 ± 1.4 | 4.2 ± 1.5 |
| LxVP (NHE1 Q678-P694) | 4300 ± 500 | −8.6 ± 1.0 | −1.3 ± 1.1 |
| *Minimal NHE1 CN-binding domain* | | | |
| NHE1ct$_{\Delta92}$ (NHE1ct I680-S723) | 21 ± 2 | −8.5 ± 0.4 | 2.0 ± 0.3 |
| NHE1ct$_{\Delta92}$ (PVIVIT) | 4.0 ± 0.4 | −9 ± 2 | 2.7 ± 2.3 |
| *Charge neutral NHE1ct dynamic linker* | | | |
| NHE1ct$_{\Delta92RARA}$ | 21 ± 2 | −12.2 ± 0.1 | −1.7 ± 1.2 |
| *TxxP mutant* | | | |
| NHE1ct P782A | 48 ± 2 | −2.8 ± 0.1 | 7.2 ± 0.1 |

All measurement with $n = 2$ or 3
[a]WT NHE1ct and all NHE1ct variant measurements performed with WT CN (CNA$_{1-370}$:CNB$_{1-170}$)
[b]Errors (±) are S.D.

**The motif-connecting linker dynamically interacts with CN.** Despite the observation that most of the NHE1 residues linking the LxVP and PxIxIT motifs exhibited line broadening (Fig. 2a, b), no electron density was detected for residues H689-E712 (Fig. 3a). Thus, these linker residues remain flexible and engage in a dynamic interaction with CN. This is further supported by the ITC data for the NHE1ct$_{ATAP}$ and NHE1ct$_{AVATAA}$ variants and their corresponding peptides that report increases in affinity from interactions of linker residues with CN (Table 1). Because the distance between the C-terminal residue of the LxVP motif and the N-terminal residue of the PxIxIT motif is too short to allow the linker to span the back side of CN, the structure also reveals that the dynamic linker connects the motifs by spanning the front, active-site containing side of CN.

Modeling potential paths that the linker may take to connect the two motif binding pockets revealed a set of basic residues in NHE1 ($^{698}$RAR$^{700}$) ideally positioned to electrostatically interact with multiple acid patches (AP) on CN (AP1:E205/D211/D229; AP2: E246; AP3: E237/D238; Fig. 3f). To determine if NHE1 residues $^{698}$RAR$^{700}$ contribute to CN binding, we generated the variant NHE1ct$_{\Delta92RARA}$ (NHE1ct$_{\Delta92}$ $^{698}$AAA$^{700}$), which lacks the ability to engage in charge:charge protein:protein interactions, and then measured its affinity for CN using ITC. While the affinity is unchanged (Supplementary Fig. 3B; Table 1), a change

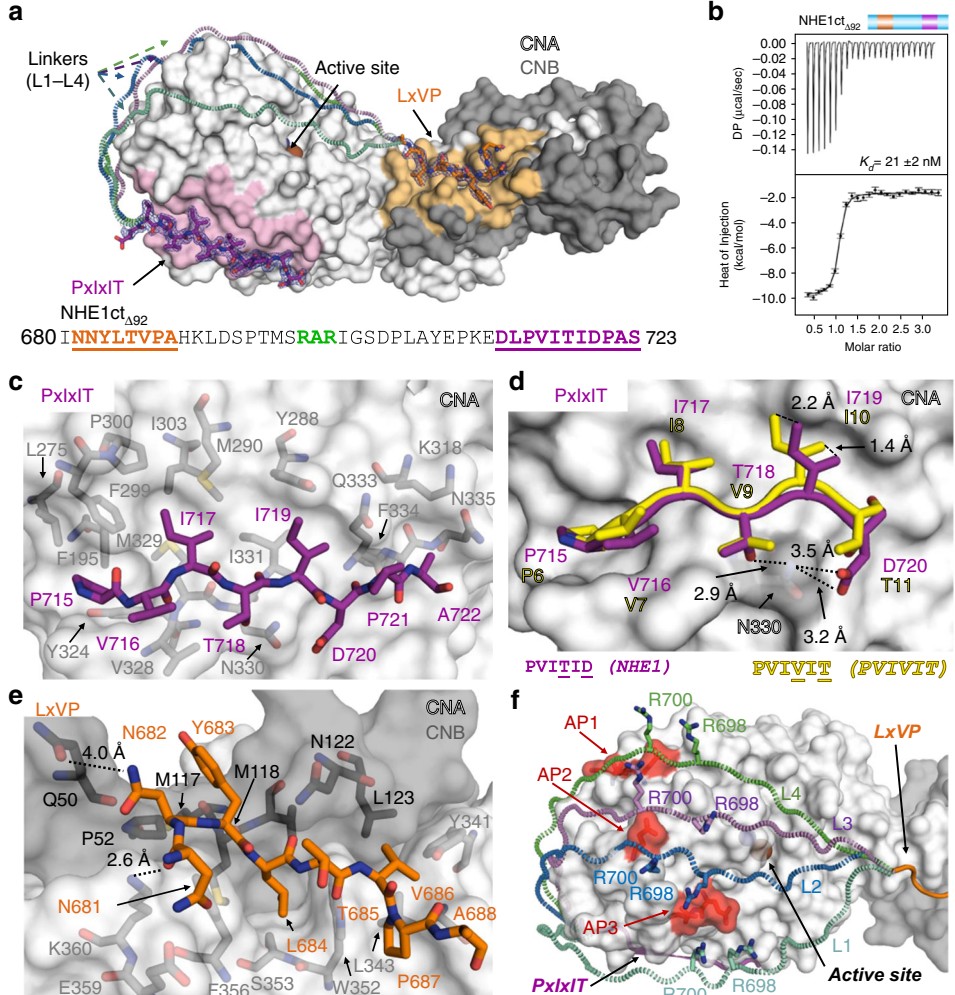

**Fig. 3** The NHE1 CN-binding motifs are connected by a dynamic linker. **a** Crystal structure of NHE1ct$_{\Delta 92}$:CN. NHE1 residues are shown as sticks in dark orange (NHE1 N681-A688, LxVP motif) or magenta (NHE1 D713-S723, PxIxIT motif); the $2F_o$–$F_c$ electron density map is contoured at $1\sigma$ (blue mesh). CN is shown as surface (CNA: white, CNB: gray, the LxVP and PxIxIT-binding pockets are shaded light orange and light pink, respectively). Bound metals are shown as spheres (Fe: brown, Zn: slate). Multiple modeled conformations of the linker are illustrated with dashed lines (L1–L4) reflecting the dynamic nature of the linker. NHE1ct$_{\Delta 92}$ sequence; residues modeled into electron density are underlined, with the LxVP and PxIxIT sequences colored as in **a**; $^{698}$RAR$^{700}$ highlighted in green. **b** ITC data for NHE1ct$_{\Delta 92}$:CN binding ($n = 2$). **c** NHE1 PxIxIT ($^{715}$PVITIDPA$^{722}$; magenta) interactions with CNA (white). NHE1 P715, I717, and I719 bind in three deep hydrophobic pockets of CNA. **d** Overlay of NHE1 residues $^{715}$PVITID$^{720}$ with that of the $^{6}$PVIVIT$^{11}$ peptide structure (yellow, PDBID: 2P6B [http://dx.doi.org/10.2210/pdb2P6B/pdb]; CNA from both structures was superimposed). NHE1 I719 rotates out of the pocket compared to the bound $^{6}$PVIVIT$^{11}$ peptide. CNA N330 is shown as sticks. NHE1 residues T718 and D720 form hydrogen bonds with CNA N330. **e** NHE1 LxVP ($^{681}$NNYLTVP$^{687}$; dark orange) interactions at the CNA/CNB (white/gray) interface. NHE1 N681 and N682 form hydrogen bonds with CNA K360 and CNB Q50, respectively. **f** Multiple modeled conformations of the linker bound to CN. Linkers shown in mint (L1), blue (L2), light purple (L3), and green (L4) dashed loops. NHE1 R698 and R700 side chains are shown as sticks while CNA is shown as surface. Acidic patches (AP1:E205/D211/D229; AP2: E246; AP3: E237/D238) are highlighted in red

in T$\Delta$S indicated reduced flexibility in the charge neutral interaction; thus, NHE1 R698 and R700 likely mediate dynamically important electrostatic interactions with CN.

To confirm these results, we used NMR spectroscopy. An overlay of the 2D [$^1$H,$^{15}$N] HSQC spectrum of $^{15}$N-labeled NHE1ct$_{\Delta 92RARA}$ in the presence and absence of CN showed that nearly all peaks broadened beyond detection (Supplementary Fig. 4). This confirms the ITC results of reduced flexibly. We also determined the 3D structure of the NHE1ct$_{\Delta 92RARA}$:CN complex to a resolution of 1.9 Å (Supplementary Fig. 5; Supplementary Table 1). Electron density is observed for CN and for NHE1 residues I681-A688 (LxVP motif) and D713-S723 (PxIxIT motif) but not for the linker; this shows that the intervening linker does not form a single conformation that can be identified using X-ray

crystallography and highlights the importance of using complementary biophysical techniques for studying dynamic interactions.

**CN specifically dephosphorylates NHE1 pT779.** Having established that NHE1 binds CN via both a PxIxIT and LxVP motif, we asked whether NHE1 is simply a CN scaffold or is additionally a CN substrate. Recently, we used NMR-based phosphorylation assays to show that the NHE1ct is phosphorylated by ERK2 at multiple sites ($p$S693, $p$S723, $p$S726, $p$S771, $p$T779, $p$S785;[11] Fig. 4a). These sites are phosphorylated similarly by other MAPKs (p38$\alpha$ and JNK1; Supplementary Fig. 6). Here, we used NMR spectroscopy to monitor CN-mediated dephosphorylation

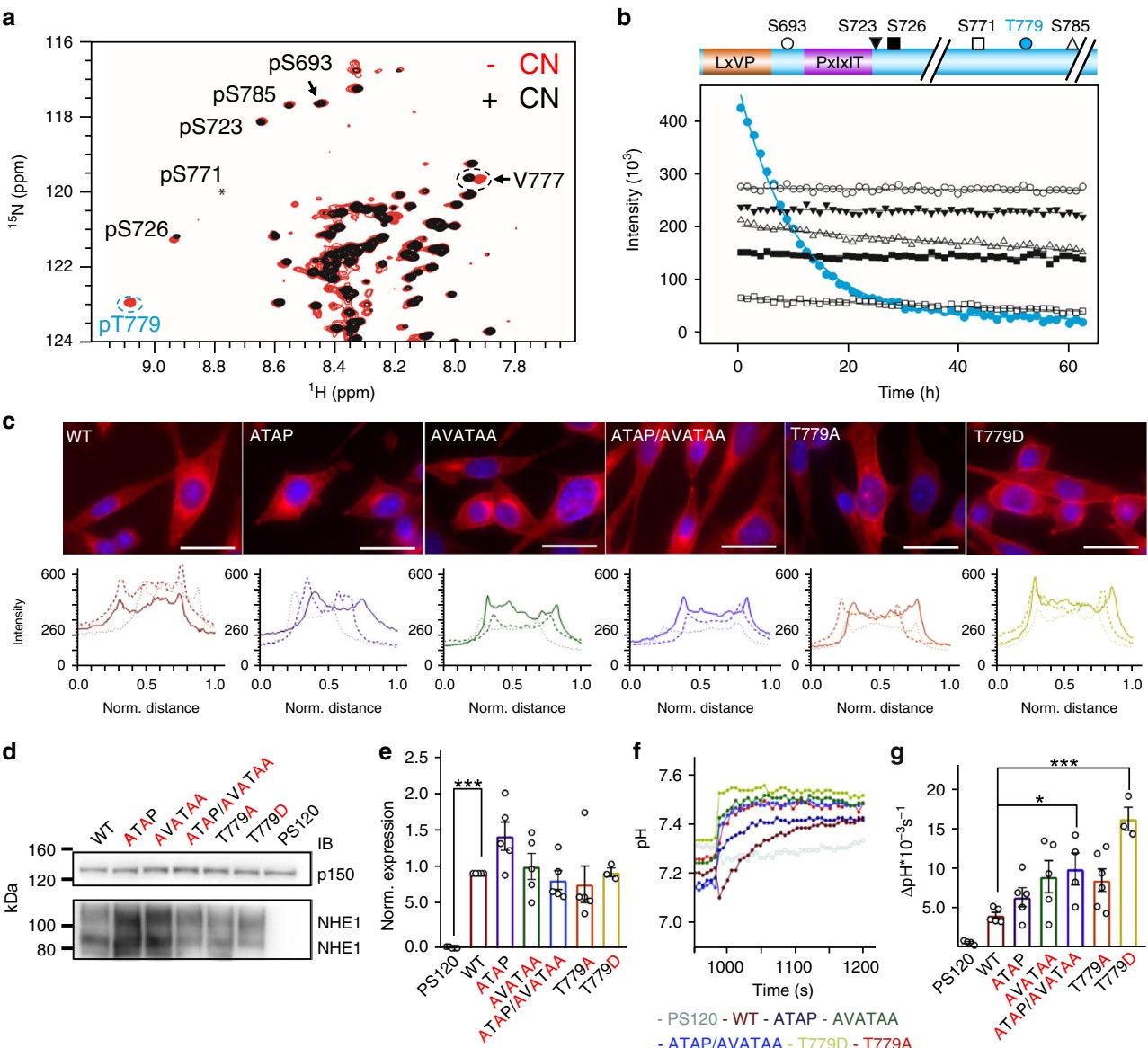

**Fig. 4** NHE1 T779 is rapidly and specifically dephosphorylated by CN. **a** Overlay of extracts from the 2D [$^1$H,$^{15}$N] HSQC spectra of ERK2-phosphorylated NHE1ct before (red), and after the addition of CN (black). Addition of CN resulted in disappearance of peaks for the phosphorylated state of T779 and reappearance of peaks reporting on the unphosphorylated state of T779 (V777, dashed circle). The peaks corresponding to the five SP phosphorylation sites were not affected. **b** Time course of in vitro NHE1ct dephosphorylation by CN following the peaks from the six ERK2 phosphorylated residues. Cartoon of NHE1ct (colored as in Fig. 1c), with the symbols of the phosphorylation sites corresponding to the dephosphorylation time-course curves shown below. **c** Top: Representative immunofluorescence (IF) images of the six variant PS120 cell lines stained for NHE1 (red) and DAPI (blue) to evaluate NHE1 expression and localization. Images were taken at ×60 magnification on an Olympus Cell Vivo microscope. Scale bars represent 20 μm (n = 2). Bottom: Representative line scans showing the membrane localization of each variant. (For overview images confirming clonality and cell surface biotinylation to confirm membrane localization see Supplementary Fig. 7.) **d** Representative Western blot of the six variant PS120 cell lines blotted for NHE1. The lower band corresponds to immature unglycosylated NHE1 and the upper band to glycosylated NHE1; p150 was used as loading control. **e** Total NHE1 expression was quantified using ImageJ and normalized for each variant to the loading control and then to the expression of the WT cell line. One-way ANOVA with Dunnet's post-test (n = 5 biologically independent experiments for all cell lines except 779, where n = 3). **f** Representative traces of BCECF-AM fluorescence converted to pH$_i$ for the six variant PS120 cell lines showed over the time course of addition of Na$^+$-free solution and Ringer solution. The recovery rates as ΔpH*10$^{-3}$*s$^{-1}$ of each cell line are depicted in **g**. One-way ANOVA with Dunnet's post-test. (The number of biologically independent experiments were as follows: T779A: 6n; WT, ATAP, AVATAA: 5n; P120, ATA/AVATAA: 4n; 779D: 3n.) * and *** denotes p < 0.05 and p < 0.001, respectively. Error bars represent S.E.M. values. Source data are provided as a Source Data file

of phosphorylated NHE1ct. Of these six ERK2 phosphorylated residues, only pT779 was specifically and rapidly dephosphorylated by CN (Fig. 4a, b; Table 2, Supplementary Table 2).

To explore if phosphorylation of T779 impacts NHE1 function, WT NHE1 and the variants NHE1 T779A (prevents phosphorylation) and NHE1 T779D (mimics phosphorylation) were

expressed in PS120 cells that lack endogenous NHE activity. All NHE1 variants localized to the plasma membrane similarly to WT NHE1 (Fig. 4c, Supplementary Fig. 7) and expressed to similar levels (Fig. 4d, e). To measure NHE1 activity, cells were loaded with the pH-sensitive fluorophore BCECF-AM, and subjected to an NH$_4$Cl prepulse-induced acidification in the

**Table 2 Apparent in vitro dephosphorylation rates of NHE1 by CN**

| NHE1ct or CN variant | Site | Phosphosite sequence[a] | $k_{dephos}$ ($10^{-3}$ $h^{-1}$)^ |
|---|---|---|---|
| *NHE1ct WT and variants*[b] | | | |
| NHE1ct | | | |
| NHE1ct WT | pT779 | $^{771}$SPGTDDVF<u>T</u>PAPSDSPS$^{787}$ | 98 ± 3 |
| *NHE1ct motif variants* | | | |
| NHE1ct$_{AVATAA}$ | pT779 | $^{771}$SPGTDDVF<u>T</u>PAPSDSPS$^{787}$ | 11 ± 3 |
| NHE1ct$_{ATAP}$ | pT779 | $^{771}$SPGTDDVF<u>T</u>PAPSDSPS$^{787}$ | 642 ± 28 |
| NHE1ct$_{AVATAA/ATAP}$ | pT779 | $^{771}$SPGTDDVF<u>T</u>PAPSDSPS$^{787}$ | 811 ± 55 |
| *NHE1ct dynamic linker variant* | | | |
| NHE1ct$_{RARA}$[c] | pT779 | $^{771}$SPGTDDVF<u>T</u>PAPSDSPS$^{787}$ | 199 ± 10 |
| *Role of Thr vs. Ser in CN-mediated dephosphorylation* | | | |
| NHE1ct T779S | pT779S | $^{771}$SPGTDDVF<u>S</u>PAPSDSPS$^{787}$ | 1 ± 2 |
| NHE1ct S785T | pT779 | $^{771}$SPGTDDVF<u>T</u>PAPSDTPS$^{787}$ | 193 ± 5 |
| | pS785T | $^{777}$VF<u>T</u>PAPSD<u>T</u>PSSQRIQ$^{792}$ | 35 ± 7 |
| *Role of TxxP motif in CN-mediated dephosphorylation* | | | |
| NHE1ct P782A | pT779 | $^{771}$SPGTDDVF<u>T</u>PAASDSPS$^{787}$ | 28 ± 3 |
| NHE1ct S785T/S787A/S788P | pT779 | $^{771}$SPGTDDVF<u>T</u>PAPSDTPA$^{787}$ | 102 ± 7 |
| $^{785}$TPAP$^{788}$ | pS785T | $^{777}$VF<u>T</u>PAPSD<u>T</u>PAPQRIQ$^{792}$ | 96 ± 7 |
| *CN variants*[d] | | | |
| CNA | pT779 | $^{771}$SPGTDDVF<u>T</u>PAPSDSPS$^{787}$ | 347 ± 10 |
| CN$_{AA}$[c] | pT779 | $^{771}$SPGTDDVF<u>T</u>PAPSDSPS$^{787}$ | 60 ± 10 |

[a]Sequence: dephosphorylated residue (underlined)
[b]All reactions performed using CN and the NHE1ct constructs indicated
[c]Reactions performed with a 20:1 (vs.100:1) NHE1ct:CN ratio and the rates normalized using wt-NHE1ct
[d]All reactions performed using NHE1ct with the CN constructs indicated
^Apparent rates of dephosphorylation were extracted from global non-linear least-square fits of disappearing peaks (dephosphorylation) to single exponentials in SigmaPlot. The standard errors (±) of the estimated parameters were reported to represent confidence intervals

absence of $HCO_3^-$. Under these conditions, recovery from acidification represents the activity of the exogenously expressed NHE1. As a control, untransfected PS120 cells showed no $pH_i$ recovery (Fig. 4f, g). Cells expressing the NHE1 T779D phosphomimetic mutant had a $pH_i$ recovery rate that was ≥3-times faster than cells expressing WT NHE1. By comparison, NHE1 T779A showed a non-significant increase in $pH_i$ recovery rate. Furthermore, cells expressing the NHE1$_{ATAP/AVATAA}$ mutant also exhibited increased $pH_i$ recovery rate compared to WT NHE1. Together, these results show that the interaction with CN, and specifically the dephosphorylation of NHE1 $p$T779, negatively regulates NHE1 activity, and conversely, that phosphorylation of T779 increases the transport activity of NHE1.

**Specificity of CN for pT779 is not due to Thr preference**. Having established that NHE1 T779 is a functionally important phosphorylation site and that its regulation by CN is essential for NHE1 function in cells, we set out to identify the molecular mechanisms by which CN specifically recognizes and dephosphorylates NHE1 $p$T779. The inability of CN to dephosphorylate $p$S693, $p$S723, and $p$S726 is explained by their close proximity to the NHE1 LxVP and PxIxIT motifs (Figs. 1c and 3a), which sterically prevents them from reaching the CN active site. However, both NHE1 $p$S771 and $p$S785, along with $p$T779, are ≥50 residues C-terminal to the NHE1 PxIxIT motif. Because all three residues can readily reach the CN active site, additional mechanisms must be responsible for restricting CN activity to only NHE1 $p$T779.

PPPs preferentially dephosphorylate Thr over Ser, although the underlying molecular mechanism is unknown[27]. Thus, one possibility for the preferred dephosphorylation of $p$T779 is that it is a Thr (Fig. 5a). To test this, we repeated the dephosphorylation experiments using a NHE1ct T779S variant. As expected, CN dephosphorylated $p$T779S more slowly than $p$T779 (WT) (Fig. 5b). However, the dephosphorylation of $p$T779S was still faster than the rate measured for any other NHE1ct phosphorylated Ser residues ($p$S693, $p$S723, $p$S726, $p$S771,

$p$S785; Fig. 4b; Table 2, Supplementary Table 2). To further confirm the specificity of CN for Thr, we also generated a NHE1ct S785T variant. CN did dephosphorylate $p$S785T; however, the rate of dephosphorylation was much slower than that observed for $p$T779 (Fig. 5c; Table 2, Supplementary Table 2). Thus, while these data confirm the preference of CN for Thr over Ser residues, generally, the enhanced dephosphorylation of $p$T779 specifically is not solely explained by this preference.

**The motifs differentially modulate NHE1 dephosphorylation**. To determine the role(s) of the PxIxIT and LxVP motifs for dephosphorylation of NHE1ct, we repeated the NMR time course experiments using the NHE1ct$_{ATAP}$, NHE1ct$_{AVATAA}$, and NHE1ct$_{ATAP/AVATAA}$ variants (Fig. 5d; Table 2, Supplementary Table 2). The data showed the following. First, abolishing the NHE1 PxIxIT:CN interaction (NHE1ct$_{AVATAA}$) reduced the dephosphorylation rate of $p$T779 (no additional sites were dephosphorylated). Second, abolishing the NHE1 LxVP:CN interaction (NHE1ct$_{ATAP}$) increased the dephosphorylation rate for $p$T779, accompanied by slow dephosphorylation of $p$S771 and $p$S785 (Table 2, Supplementary Table 2). Third, dephosphorylation of NHE1ct$_{ATAP/AVATAA}$ resembled that of NHE1ct$_{ATAP}$ with additional slow dephosphorylation of $p$S693, $p$S723, and $p$S726 (Table 2, Supplementary Table 2). Fourth, to test the contribution of the LxVP interaction, we repeated the dephosphorylation of $p$T779 using only CNA, which lacks the LxVP-binding pocket (this pocket is only present when CNB is bound to CNA via CNA residues 348–370). Again, the dephosphorylation rate of $p$T779 was significantly faster (Fig. 5e; Table 2, Supplementary Table 2). Finally, we tested the contribution from electrostatics and thereby the dynamics of the 27-residue linker using NHE1ct$_{RARA}$ (NHE1 $^{698}$AAA$^{700}$), which led to a 1.5-fold increased rate of $p$T779 dephosphorylation (Supplementary Fig. 8A; Table 2). This shows that the 27-residue linker directly negatively influences the rate of $p$T779 dephosphorylation.

These dephosphorylation results are fully consistent with our molecular data. First, loss of binding at the PxIxIT site (CN with

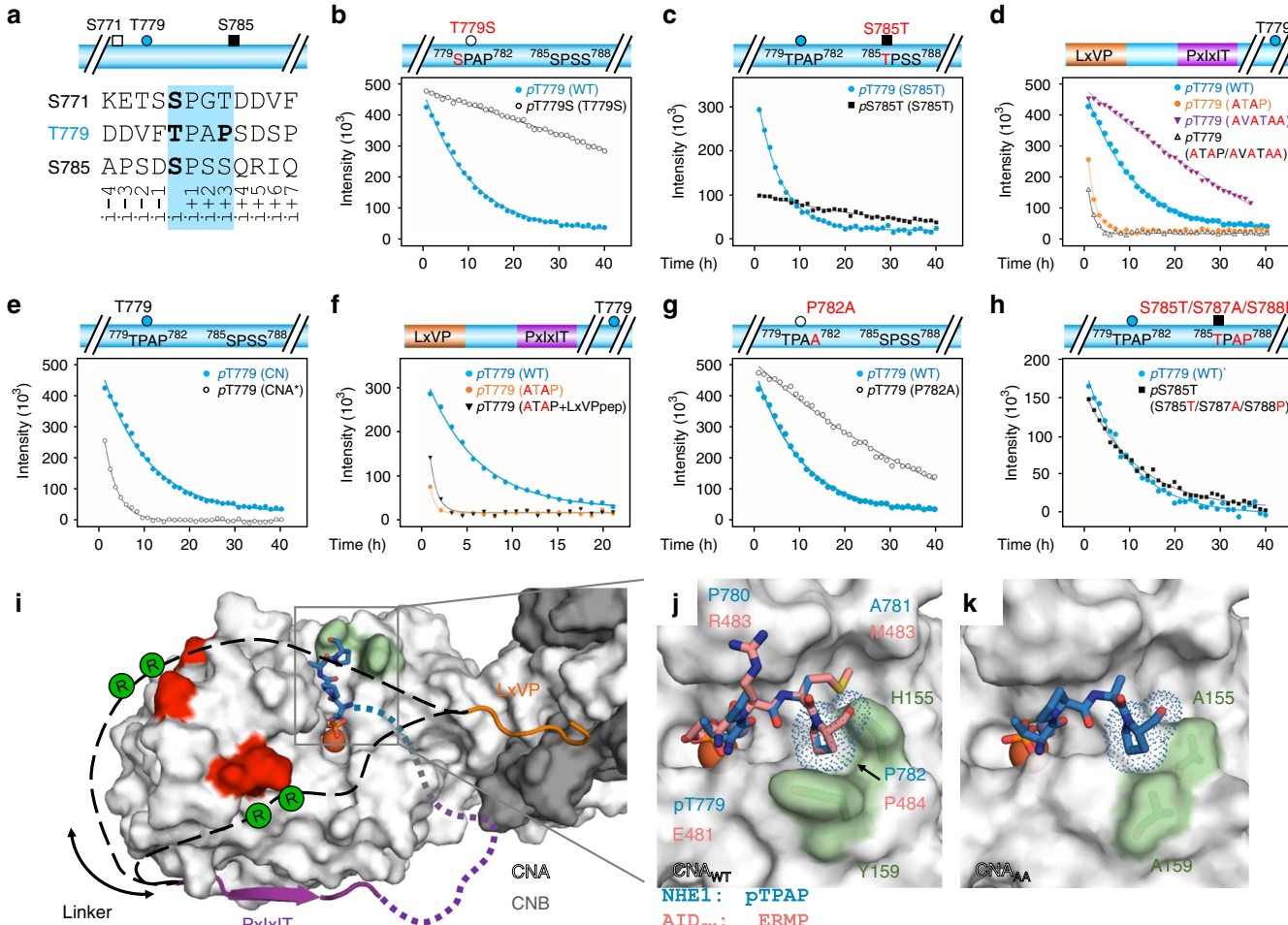

**Fig. 5** Role of docking motifs for NHE1 dephosphorylation and identification of a TxxP motif. **a** Sequences of the three phosphosites capable of reaching the CN active site in WT NHE1ct: pS771 (white box), pT779 (blue ball), and pS785 (black box); phosphorylated residues in bold as is the P of the T779 TxxP motif. **b** Dephosphorylation of NHE1ct WT and T779S (monitoring pT779 and pT779S, respectively). **c** Dephosphorylation of NHE1ct S785T (monitoring pT779 and pS785T). **d** Dephosphorylation of NHE1ct WT and docking motif variants (NHE1ct_ATAP, NHE1ct_AVATAA, and NHE1ct_ATAP/AVATAA; monitoring pT779). **e** Dephosphorylation of WT NHE1ct by CN (CNA/CNB) or CNA* (monitoring pT779). **f** Dephosphorylation of WT NHE1ct, NHE1ct ATAP, and NHE1ct ATAP in the presence of saturation concentrations of LxVP peptide (monitoring pT779). **g** Dephosphorylation of NHE1ct WT and P782A (monitoring pT779). **h** Dephosphorylation of NHE1ct triple mutant S785T/S787A/S788P (monitoring pT779 and pS785T). **i** Model for NHE1 substrate binding to CN. CN shown as surface (CNA: white, CNB: gray); acidic patches in red. NHE1 residues are shown in dark orange (NHE1 N681-A688, LxVP motif) or magenta (NHE1 D713-S723, PxIxIT motif) with the dynamic linker between the PxIxIT and LxVP motifs shown as dashed lines (black). The locations of NHE1 residues R698 and R700 are indicated by green circles. CN active site includes a modeled pTPAP peptide (blue). **j** Overlay of CN AID [481]ERMP[484] (PDBID 1AUI) and a modeled pTPAP sequence at the CN active site, highlighting the i+3 Pro binding pocket on CN (light green; H155/Y159). **k** NHE1 NHE1ct_Δ92:CN_AA structure with the modeled pTPAP peptide from **i**, **j**. Source data are provided as a Source Data file

NHE1ct_AVATAA) lead to an overall reduction in the dephosphorylation rate of pT779 without loss of specificity, most easily explained by the increased distance between the substrate site (pT779) and the closest docking site on CN for this variant, i.e. the LxVP motif. In contrast, loss of binding at the LxVP site (CN with NHE1ct_ATAP or NHE1ct_ATAP/AVATAA; NHE1ct with CNA) negated the unique specificity for pT779. This effect most likely stems from an abolished interaction between CN and the 27-residue linker that connects the NHE1 LxVP and PxIxIT-binding motifs. As established, this linker interacts in a dynamic manner with CN. To confirm this model, we repeated the dephosphorylation of NHE1ct_ATAP by CN in the presence of a large excess of LxVP peptide (CN:NHE1ct_ATAP: LxVP_peptide molar ratio of 1:100:100). Based on the respective concentrations and affinities, this should result in nearly 100% occupancies of both the LxVP and PxIxIT-binding sites (by the LxVP_peptide and NHE1ct_ATAP, respectively). Again, a much faster dephosphorylation rate was observed (Fig. 5f; Table 2), further

supporting that the 27-residue linker connecting the LxVP and PxIxIT sites restricts access to the CN active site and that anchoring the linker via the LxVP CN pocket controls this specificity.

Our dephosphorylation data show that the presence of the linker not only reduces the dephosphorylation rate of pT779 but also increases the specificity of CN for only pT779. This is because slow dephosphorylation of pS771 and pS785 is only observed in the absence of linker-anchoring at the LxVP site (i.e. with NHE1ct_ATAP). Considering the difference in co-localization, these results are consistent with an observed increase in acidification recovery in NHE1 motif variants compared to WT NHE1 when expressed in PS120 cells (Fig. 4f, g).

**A TxxP motif is a preferred substrate for CN.** While the NHE1 linker influences CN dephosphorylation specificity and rates, an activity that requires the LxVP interaction, these interactions are

not solely responsible for the observed specificity of CN for NHE1 $p$T779. We reasoned that residues surrounding T779 also play a role. The only structural information describing interactions at the CN active site is from the structure of CN bound to the CN AID ($^{481}$ERMP$^{484}$)[19]. AID residue E481, which mimics a phosphorylated residue, binds at the active site, blocking substrate access. The AID sequence has a proline residue in the i + 3 position (P484). This is of interest because phosphorylated TxxP motifs identified in CN regulators and substrates have been hypothesized to bind directly at the CN active site[28].

NHE1 T779 is also part of a TxxP motif ($^{779}$TPAP$^{782}$). This led us to hypothesize that NHE1 P782 contributes to the specific dephosphorylation of $p$T779 by CN. Using ITC, we showed that the affinity of the NHE1ct$_{P782A}$ variant for CN was unchanged (Supplementary Fig. 3C; Table 1). In spite of the identical affinity, the dephosphorylation of $p$T779 was impaired in this variant (Fig. 5g; Table 2), demonstrating the proline in the i + 3 position (P782) is essential for robust dephosphorylation. To confirm the role of the TxxP motif as a CN-specific substrate determinant, an additional TxxP motif was introduced at a different NHE1ct site. Specifically, the NHE1 S785 sequence was replaced with a TxxP sequence ($^{784}$DSPSSQR$^{790}$ to DTPAPQR) and the dephosphorylation analysis repeated (Fig. 5h). The dephosphorylation rates for the Thr residues in both TPAP motifs (T779 and S785T, both TxxP) were fast and similar (Table 2, Supplementary Table 2), confirming the role of the TxxP motif as a bona fide substrate determinant for CN.

**Molecular basis for TxxP recognition.** Next, we created a model where we replaced the CN AID $^{481}$ERMP$^{484}$ sequence with the NHE1 $^{779}$TPAP$^{782}$ sequence (Fig. 5i). We speculated that NHE1 P782 likely binds in the same hydrophobic pocket as that used by CN AID P484; namely, that created by CN H155 and Y159 (Fig. 5j). To test this, we generated the CN$_{AA}$ variant (CNA: H155A/Y159A, CNB). CN and CN$_{AA}$ had identical melting temperatures, confirming the maintenance of overall structure (Supplementary Table 3). We also measured CN activity using $p$-nitrophenyl phosphate (pNPP; a small, general none-specific substrate for CN) assays. Both CN and CN$_{AA}$ had identical $K_m$ and $k_{cat}$ values, showing that catalysis is identical for both CN variants (Supplementary Table 3). We then determined the structure of the NHE1ct$_{\Delta92}$:CN$_{AA}$ complex (2.3 Å resolution; Supplementary Fig. 5B; Supplementary Table 1). As expected, electron density was observed for CN$_{AA}$ and for NHE1 residues N682-A688 (LxVP motif) and D713-S723 (PxIxIT motif) (Supplementary Fig. 5C). However, the NHE1ct$_{\Delta92}$:CN$_{AA}$ structure had a much shallower H155A/Y159A Pro-binding pocket than WT CN, while the remaining structure was unchanged (Fig. 5k, Supplementary Fig. 5B). Indeed, comparing the buried surface area of a modeled TPAP sequence for CN and CN$_{AA}$ showed the NHE1 P782 pocket to be ~36% larger in CN. We then measured the dephosphorylation rate of $p$T779 by CN$_{AA}$, which decreased by ~50% compared to that with WT CN (Supplementary Fig. 8B; Table 2, Supplementary Table 2). This strongly supports the conclusion that the NHE1 TxxP sequence interacts with CN in a manner similar to the CN AID. Thus, the presence of a Pro at the i + 3 position (relative to $p$Thr) enhances a substrate's affinity for the CN active site by interacting with the H155/Y159 pocket. Because these residues are not conserved between the family of PPPs, the observed i + 3 Pro selectivity is specific for CN (Supplementary Fig. 5D).

## Discussion

The balance between phosphorylation and dephosphorylation controls cell signaling and cellular communication. Ser/Thr

kinases, the enzymes catalyzing phosphorylation, use protein:protein interactions and substrate-specific recognition sequences to identify their substrates. Conversely, the mode(s) of substrate recognition for the PPP Ser/Thr protein phosphatase family (PP1, PP2A, PP2B/3, PP4, PP5, PP6, and PP7) is not known. While it has become clear that at least PP1, PP2A, PP2B, and PP4 utilize protein:protein interactions for substrate binding[24,26,29], it is still unclear if and how these interactions direct substrate specificity for single dephosphorylation sites, often 50 residues or more away from the scaffolding motifs. Our lack of understanding of molecular selectivity determinants in PPPs hampers bioinformatics analysis to assign PPP:substrate pairs, thus impairing progress in obtaining a systems biology understanding of signaling.

One protein that is tightly regulated by both protein:protein interactions and post-translational modifications is NHE1, an evolutionarily conserved acid-extruding transporter serving as a key regulator of cellular pH$_i$ in essentially all mammalian cells studied[4]. Six NHE1ct residues are phosphorylated by ERK2, a kinase that is often activated downstream of, or in parallel with, increases in [Ca$^{2+}$]$_i$[30,31]. A previous report showed that an NHE1 T779A/S785A mutant exhibited a tendency for reduced NHE1 activation by sustained acidosis[32]. We show here that phosphorylation of T779 increases NHE1 activity and, in turn, $p$T779 is dephosphorylated by the Ca$^{2+}$-activated phosphatase, CN. This suggests the existence of a feedback mechanism, in which the extent and duration of NHE1 activation by ERK are restricted by its dephosphorylation following [Ca$^{2+}$]$_i$-induced CN activation. As excessive NHE1 activation is associated with, e.g. uncontrolled cell division and growth[3], this intrinsic brake on its activity is likely physiologically meaningful. In conjunction with previous work showing that CN activity in vivo is regulated by NHE1-dependent pH$_i$ changes[16], this reveals the existence of a dynamic complex of mutual regulatory interactions with the NHE1–CN interaction at its core.

We show that the two canonical CN-specific SLiMs, a previously identified PxIxIT motif and a newly discovered LxVP motif, contribute to NHE1 binding of CN. As observed for other CN regulators and inhibitors, minor changes in these motifs lead to significant changes in interaction strength[22,26]. Interestingly, mutation of both sites was insufficient to abolish endogenous CN binding to full-length NHE1 in a cellular context—likely reflecting the known, extensive scaffolding properties of NHE1, which forms dynamic, multiprotein complexes at the plasma membrane[33]. Further, in NHE1, the LxVP sequence ($^{684}$LTVP$^{687}$) is not only an LxVP motif but also functions as a D-domain/Kinase Interaction Motif (KIM) that is essential for ERK2 binding and scaffolding. The use of an overlapping sequence ensures that ERK2 and CN binding to NHE1 is mutually exclusive.

We also discovered that the LxVP motif is essential for directing the specificity of CN for NHE1 $p$T779. However, it does so in an unanticipated and indirect manner. Namely, NHE1 forms a critical anchor for a 27-residue linker that dynamically associates with CN across its front surface. In doing so, it reduces access to the CN active site, which in turn, reduces the overall speed of dephosphorylation. Simultaneously, this increases the specificity towards a single phosphorylated residue, $p$T779. While it has been known for more than 30 years that CN prefers $p$Thr over $p$Ser substrates[27], we discovered that this preference alone does not fully explain selectivity for pT779. Rather, a proline in the i + 3 position (relative to $p$Thr, TxxP) makes a specific and unique interaction with a CN surface binding pocket formed by CN residues H155 and Y159. These residues are not conserved in any other PPP, i.e., the specific dephosphorylation of the TxxP motif is unique to CN. Taken together this shows that CN uses a combination of mechanisms, including a dynamic interaction

 9

with a 27-residue linker that restricts substrate access to a unique interaction pocket adjacent to the catalytic site leading to a preference for TxxP motifs, to achieve exquisite specificity towards a single Thr in NHE1, resulting in rapid and specific dephosphorylation of this residue, hence limiting the duration of NHE1 activation following its ERK-induced phosphorylation. This is furthermore a remarkable example on how disordered linkers in complexes have direct functional relevance, in this case acting as a substrate specificity filter for CN.

While this is clearly a complex mechanism, a number of aspects will be true for other CN substrates. Current data show that the CN interaction SLiMs (PxIxIT and LxVP) have variable roles in substrate recruitment and only minor roles for specificity. In contrast, the i + 3 proline-binding pocket in CN allows for specific recruitment to the active site. Thus, identification of the PxIxIT and/or LxVP motifs, together with an [S/T]xxP site provides a powerful computational strategy for identifying and testing currently unrecognized CN substrates. To support this notion, we used our previously developed CNcon database (48 experimentally confirmed CN interaction proteins)[26]. Thirty-three CNcon members were previously shown to contain one or more LxVP sites (11 of which have been experimentally verified). Here, we extended the analysis and searched for [ST]xxP sequences in IDRs. We discovered that 32 CNcon members have confirmed/predicted LxVP and [ST]xxP sites in an IDR, with 50% of the [ST]xxP sites positioned 10–35 residues from the LxVP site (C-terminal) and ~90% within 10–100 residues (≥9 residues are required to bridge the CN LxVP-binding pocket to the CN active site). The number of [ST]xxP sites in a single substrate is highly variable (1–42). Further, when examined together, the identified [ST]xxP sites have an equal distribution of Ser/Thr residues. Although not every identified [ST]xxP site is likely a bona fide CN substrate, our data suggests that many will be. Thus, depending on the phosphorylating kinase and distinct spatial, temporal, and cellular inputs, multiple different [S/T]xxP sites within a single substrate may barcode time-dependent CN activity.

Together, this study not only highlights the importance of T779 phosphorylation for NHE1 function and regulation, but provides the most comprehensive molecular understanding of how CN recruits a specific substrate. First, our data suggest the existence of a feedback mechanism in which the extent and duration of NHE1 activation by ERK are restricted by its dephosphorylation following $[Ca^{2+}]_i$-induced CN activation. Second, our data identify an active site recognition sequence for CN, TxxP. These results are in striking contrast to the recruitment of substrates to PP1, which uses steric exclusion and multi-domain scaffolding. Thus, this specific amino acid sequence ([S/T]xxP) can now be leveraged—together with PxIxIT and/or LxVP motifs—to identify and test specific CN substrates sites. Because of the importance of CN activity for a diversity of biological functions, our discoveries not only provide key insights into NHE1 function and regulation, but also provide fundamental advances for establishing a systems-level understanding of CN signaling networks.

## Methods

**Sequence alignments.** ClustalW was used to create all sequence alignments. The following species are used in Fig. 1c: *Amphiuma tridactylum* (atNHE1; salamander), *Xenopus laevis* (xlNHE1; clawed frog), *Mus musculus* (mmNHE1; mouse), *Rattus norvegicus* (rnNHE1; rat), *Cricetulus griseus* (cgNHE1; Chinese hamster); *Sus scrofa* (ssNHE1; pig), *Bos taurus* (btNHE1; cow), *Homo sapiens* (hNHE1; human), *Oryctolagus cuniculus* (ocNHE1; rabbit), *Gallus gallus* (ggNHE1; chicken), *Takifugu rubripes* (trNHE1; Japanese pufferfish), *Pleuronectes americanus* (paNHE1; winter flounder), *Platichthys flesus* (pfNHE1; European flounder); *Oncorhynchus mykiss* (omβNHE1; rainbow trout), and *Cyprinus carpio* (ccNHE1; carp).

**Protein expression and purification.** DNA coding the human NHE1 I680-Q815 (NHE1ct) was subcloned into a pET-11a expression vector[11]. DNA coding the human NHE1ct$_{\Delta 92}$ was subcloned into a pET-M30-MBP vector, which encodes an N-terminal His$_6$-tag followed by maltose-binding protein (MBP) and a TEV (tobacco etch virus) protease cleavage site. The following human NHE1 (see also Supplementary Fig. 1) and human CN variants were generated using the Quik-change (Agilent) site-directed mutagenesis kit: NHE1/NHE1ct$_{ATAP}$ (L684A, V686A), NHE1/NHE1ct$_{AVATAA}$ (P715A, I717A, I719A, D720A), NHE1/NHE1ct$_{ATAP/AVATAA}$ (L684A, V686A, P715A, I717A, I719A, D720A), NHE1 T779A, NHE1 T779D, NHE1ct T779S, NHE1ct S785T, NHE1ct P782A, NHE1ct S785-TPAP (S785T, S787A, S788P), NHE1/NHE1ct$_{RARA}$ (R698A, R700A), NHE1ct$_{\Delta 92PVIVIT}$, CN$_{AA}$ (alpha isoform, subunit A: M1-A391, H155A, Y159A, subunit B: 1–170), CNA/CNB (alpha isoform, subunit A: M1-N370, H155A, Y159A, subunit B: 16–170), CNA (alpha isoform: H27-D348; two C-terminal ASP residues were added for increased CNA solubility).

For protein expression, NHE1ct and CN were expressed as described[11,21]. Briefly, proteins were expressed in *E. coli* BL21 (DE3) cells (Agilent). Cell cultures were grown at 37 °C under vigorous shaking (250 rpm) to an OD$_{600}$ of 0.7. Cells were cooled at 4 °C for one hour, while the shaker temperature was lowered to 18 °C. Expression was induced by addition of 1 mM IPTG, and the cultures were grown for an additional 18 h at 18 °C (250 rpm). The cells were harvested by centrifugation (6000× *g*, 15 min, 4 °C) and stored at −80 °C. NHE1ct$_{\Delta 92}$ was expressed in *E. coli* BL21 (DE3) CodonPlus-RIL cells (Agilent). Cells were grown in Luria Broth in the presence of selective antibiotics at 37 °C up to OD$_{600}$ of 0.6–0.8, and expression was induced by the addition of 1 mM isopropyl-β-D-1-thiogalactopyranoside (IPTG). NHE1ct$_{\Delta 92}$ was expressed for 4 h at 37 °C and CN expressed for 18 h at 18 °C prior to harvesting by centrifugation at 6000 × *g*. Cell pellets were stored at −80 °C until purification. For NMR measurements, expression of uniformly $^{15}$N-labeled and/or $^{13}$C-labeled NHE1 was achieved by growing cells in M9 minimal media containing 1 g/L $^{15}$NH$_4$Cl and/or 4 g/L [$^{13}$C]-D-glucose (CIL or Isotec) as the sole nitrogen and carbon sources, respectively.

CN and NHE1ct were purified as described[11,21] or carried out as follows. For all proteins, cell pellets were lysed in lysis buffer (20 mM Tris–HCl pH 8.0, 500 mM NaCl, 5 mM Imidazole, 0.1% Triton X-100) containing EDTA-free protease inhibitor cocktail (Roche) using high-pressure homogenization (Avestin). The lysate was clarified by centrifugation at 45,000 × *g* for 60 min at 4 °C. The supernatant was filtered through a 0.22 μm filter before loading onto a 5 ml HisTrap HP column (GE Healthcare) equilibrated in Buffer A (50 mM Tris–HCl pH 8.0, 500 mM NaCl, and 5 mM imidazole). Bound proteins were washed with Buffer A and were eluted using a linear gradient of Buffer B (50 mM Tris–HCl pH 8.0, 500 mM NaCl, 500 mM imidazole). Peak fractions were pooled and dialyzed overnight at 4 °C (50 mM Tris–HCl pH 8.0, 500 mM NaCl, 0.5 mM TCEP) with TEV protease to cleave the His$_6$-tag. The next day, the protein was incubated with Ni$^{2+}$-NTA resin (GE Healthcare) to remove TEV and the cleaved His-tag. Cleaved NHE1ct or NHE1ct$_{\Delta 92}$ was incubated at 80 °C and centrifuged at 14,000 × *g* for 10 min, the supernatant was purified using size-exclusion chromatography (SEC; Superdex 75 26/60 [GE Healthcare]) equilibrated in SEC buffer (20 mM Tris–HCl pH 7.5, 100 mM NaCl, 1 mM CaCl$_2$, and 0.5 mM TCEP) and concentrated to 1–2 mg/ml prior to storage at −80 °C. CN was purified identically, except the cleaved product was further purified by anion exchange chromatography (HiTrap QHP; low salt loading buffer: 20 mM Tris–HCl pH 7.5, 50 mM NaCl, 1 mM CaCl$_2$, and 0.5 mM TCEP; high salt elution buffer: 20 mM Tris–HCl pH 7.5, 800 mM NaCl, 1 mM CaCl$_2$, and 0.5 mM TCEP) prior to SEC (Superdex 75 26/60 [GE Healthcare], equilibrated in SEC buffer; 20 mM Tris–HCl pH 7.5, 100 mM NaCl, 1 mM CaCl$_2$, and 0.5 mM TCEP).

To form NHE1ct:CN complexes, CN (CNA$_{1–370}$, CNB$_{16–170}$) was purified as described above except fractions containing CN after the anion exchange chromatography step were combined and incubated with a three-fold molar excess of NHE1ct$_{\Delta 92}$ or NHE1ct$_{\Delta 92RARA}$. The complex was then concentrated and purified using SEC (Superdex 75 26/60) pre-equilibrated with the crystallization buffer (20 mM Tris–HCl pH 7.5, 100 mM NaCl, 1 mM CaCl$_2$ and 0.5 mM TCEP).

**NMR spectroscopy.** NMR data were acquired at 25 °C on a Varian INOVA 800 MHz ($^1$H) spectrometer with a room temperature probe or a Bruker AVANCE/NEO 600 or 750 MHz ($^1$H) spectrometer equipped with cryogenic probes. Free induction decays were transformed and visualized in NMRPipe[34] or Topspin 4.05 (Bruker BioSpin) and the spectra analyzed using both the CCPN Analysis software[35] and CARA[36] (http://www.cara.nmr.ch). Interaction and time course experiments were recorded on Varian 800 MHz ($^1$H) NMR spectrometers with a 5 mm triple resonance probe with a Z-field gradient or a Bruker NEO 600 MHz ($^1$H) NMR spectrometer with a 5 mm TCI-active HCN z-gradient cryoprobe at 5 °C (interaction) or 25 °C (time-course experiments). Chemical shift referencing were relative to 4,4-dimethyl-4-silapentane-1-sulfonic acid (DSS) and spectra were zero filled, Fourier transformed and baseline corrected in NMRDraw[34] or Topspin 4.05 (Bruker BioSpin) and analyzed in CCPN Analysis[35]. For the phosphorylation experiments using different MAPKs, NMR samples of 400 μL of 200 μM $^{15}$N-labeled NHE1ct were prepared in PBS buffer pH 7.0, 5 mM ATP, 10 mM MgCl$_2$, 0.01% (w/v) NaN$_3$, 0.5 mM DSS, 5 mM DTT, 10% (v/v) 99.96% D$_2$O. 2D [$^1$H,$^{15}$N] HSQC spectra were recorded before and after incubation with the active kinases (purchased from proteinkinase.de) for 48 h at 25 °C (1 μg of active ERK2,

386 pmol/(μg*min); 2.7 μg of active p38α, 150 pmol/(μg*min); 2 μg of active JNK1, 514 pmol/(μg*min)). For the dephosphorylation experiments, a 400 μL sample of 200 μM $^{15}$N-labeled NHE1ct (WT and variants), phosphorylated by active ERK2 at all six consensus sites, was prepared in PBS buffer pH 7.2 supplemented with 10 mM MgCl$_2$, 0.5 mM CaCl$_2$, 5 mM DTT, 0.5 mM DSS, 5–10% (v/v) 99.96% D$_2$O. A reference 2D [$^1$H,$^{15}$N] HSQC spectrum was recorded before addition of phosphatase. Dephosphorylation was initiated by addition of 2 μM unlabeled active CNA or CN resulting in a molar excess of NHE1ct:CN of 100:1. For the CN$_{AA}$ and NHE1ct$_{RARA}$ variants, a 550 μL sample of 100 μM $^{15}$N-labeled NHE1ct, phosphorylated by active ERK2[37] at all six consensus sites, was prepared in 20 mM HEPES pH 6.8, 100 mM NaCl, 1 mM CaCl$_2$, 0.5 mM TCEP, 10 mM MgCl$_2$, 10% D$_2$O. A reference 2D [$^1$H,$^{15}$N] HSQC spectrum was recorded before addition of phosphatase. Dephosphorylation was started by addition of 5 μM unlabeled active CN resulting in a molar excess of NHE1ct:CN of 20:1. Dephosphorylation was monitored by extraction of intensities from a series of 2D [$^1$H,$^{15}$N] HSQC spectra. Apparent rates of dephosphorylation were extracted from global non-linear least-square fits of disappearing phosphorylated peaks and/or reporting neighbor peaks to single exponentials in SigmaPlot.

**ITC measurements**. Protein concentration of CN (CNA M1-N370, CNB M1-V170) and NHE1 variants were measured in triplicate using the Pierce 660 assay (Thermo Fisher). All protein samples were equilibrated in 20 mM Tris–HCl pH 7.5, 100 mM NaCl, 1 mM CaCl$_2$, and 0.5 mM TCEP. NHE1 or variants (30–50 μM, syringe) was titrated into CN (5–10 μM, cell) using a 250 or 300 s interval at 25 °C. Twenty-five injections were delivered during each experiment over a period of 20 s (VP-ITC microcalorimeter) or 10 s (Affinity-ITC microcalorimeter) and the solution in the sample cell was stirred at 307 rpm (VP-ITC microcalorimeter) or 125–200 rpm (Affinity-ITC microcalorimeter) to ensure rapid mixing. All data was analyzed using NITPIC and fitted to a single site-binding model using SED-PHAT[38]; figures generated using GUSSI[39].

**Proximity ligation assays**. Proximity ligation assay was carried out using a Duolink II Detection Reagents Red kit (Sigma Aldrich). WT MCF7 cells were seeded on coverslips the day before assaying (MCF-7 cells were a kind gift from Lone Ronnov-Jessen [University of Copenhagen, Denmark]). Cells were washed in ice-cold PBS, fixed in 4% PFA for 30 min on ice, and washed in PBS. Cells were quenched using 0.1 M glycine for 15 min followed by permeabilization in 0.5% Triton X-100 for 15 min. Cells were subsequently washed in Duolink II Buffer A and blocked using O-link-blocking solution for 30 min. Cells were incubated with the primary antibodies (NHE1: XB-17, 1:100 (gift from Mark Musch, University of Chicago); CN: Millipore-Sigma, #C1956, 1:200) of interest diluted in Duolink II antibody diluent for 60 min in a humidity chamber at 37 °C, and then incubated with PLA probes diluted 1:5 in Duolink II Antibody Diluent buffer at 37 °C for 60 min. Coverslips were washed in Duolink II Buffer A and incubated with ligase diluted 1:40 in 1x Duolink II ligation solution for 30 min at 37 °C, followed by wash in Duolink II Buffer A. Subsequently, coverslips were incubated with a polymerase diluted 1:80 in 1x Duolink II amplification solution for 100 min at 37 °C. After amplification, coverslips were washed in Duolink Buffer A, incubated with phalloidin488 for 1 h, and treated with DAPI to stain nuclei. Finally, coverslips were washed in Duolink Buffer A and MQ, mounted on object glass with mounting buffer, and sealed with nail polish. Imaging was carried out with an Olympus BX-61 epifluorescence microscope using cellSens Dimensions V1.6 software. Images were taken as z-stacks and z-projection images were created. Further image processing and quantification was carried out in ImageJ[40]. PLA signal was quantified using particle analysis in Image J for a minimum of 100 cells, for both control (using only anti-CN primary antibody; NHE1: XB-17, 1:100 (gift from Mark Musch, University of Chicago); CN: Millipore-Sigma, #C1956, 1:200) and NHE1-CN PLA signal. Furthermore, z-projection images were deconvoluted using the cellSens Dimension V1.6 software for high-resolution qualitative images.

**Co-Immunoprecipitation**. Immunoprecipitation (IP) was carried out for fully confluent PS120 cells, expressing WT[41], variant NHE1 or empty vector (PS120 cells were generated in the laboratory of Dr. Jacques Pouyssegur [University of Nice, France] and were a kind gift from Prof. Laurent Counillon [University of Nice Sophia Antipolis, France]). Prior to IP, cells were subjected to a 10 min incubation in physiological saline (Ringer solution, 135 mM salt) in the absence or presence of 5 μM ionomycin to increase cellular [Ca$^{2+}$]. Cells were washed with PBS and lysed in pre-heated IGEPAL lysis buffer (1% (v/v) IGEPAL CA-630, 1 mM NaF, 3 mM Na$_3$VO$_4$, EDTA-free Complete$^{TM}$ protease inhibitor, 50 mM Tris pH 7.4, and 140 mM NaCl), detached using a rubber policeman and incubated on ice for 30 min. Preparation of lysates and determination of protein contents was done as described for immunoblotting. 2 mg protein for each IP were incubated with primary antibodies against CN and NHE1, or IgG control (1 μg antibody per mg lysate) for 30 min at 4 °C with gentle rolling. 50 μl (1.5 mg) pre-washed (2 × 10 min in lysis buffer, 4 °C) Dynabeads Protein G (Invitrogen) were added to each sample, followed by incubation for 30 min at 4 °C with gentle rotation. Beads were pelleted by magnetic rack, and dynabeads with bound protein were washed 5 × 5 min in 500 μL lysis buffer, boiled for 5 min at 95 °C with 80 μL sample buffer, mixed thoroughly, and incubated on ice for 30 min to ensure full elution. Proteins were separated using SDS–PAGE and analyzed by Western blotting as above. Input samples were analyzed in parallel.

**Immunoblotting**. Cells were grown to ~80% confluence in 10 cm Petri dishes, washed in ice-cold PBS, lysed in boiling lysis buffer (1% SDS, 10 mM Tris–HCl, pH 7.5), sonicated, and centrifuged to clear debris. Identical amounts of protein (15–25 μg/well) diluted in NuPAGE LDS sample buffer (Lifetech Technologies) were boiled for 5 min, separated on NuPAGE 10% bis–tris gels, and transferred to nitrocellulose membranes using the Novex gel system (Novex, San Diego, CA). Membranes were stained with Ponceau S to confirm equal loading, blocked for 1 h at 37 °C in blocking buffer (120 mM NaCl, 10 mM Tris–HCl, 5% nonfat dry milk), and incubated with the relevant primary antibodies in blocking buffer overnight at 4 °C. After washing in TBST (TBS + 0.1% Tween-20), membranes were incubated with HRP-conjugated secondary antibodies (1:2000, Sigma), washed in TBST, and visualized using BCIP/NBT. For quantifications, blots were scanned, and band intensities quantified using Un-Scan-IT Graph Digitizer software (Silk Scientific). Protein bands were normalized to those of the loading control (p150) from the same gel to eliminate gel-to-gel differences. For immunoprecipitation samples, protein bands were normalized to the input, from the same gel. Uncropped and unprocessed scans of the blots are provided in the Source Data file. Primers and recombinant DNA used in this study are summarized in Supplementary Table 4.

**Immunofluorescence analysis of NHE1 and CN**. For immunofluorescence experiments, cells were grown on 12 mm round glass coverslips until ~80% confluency and fixed in 2% MeOH (15 min on ice). Coverslips were washed three times for 5 min in PBS, permeabilized for 15 min (0.5% Triton X-100 in TBS), blocked for 30 min (5% BSA in TBST), and incubated with primary antibody against NHE1 (antibody details see below) in TBST + 1% BSA at RT for 1.5 h. Coverslips were again washed in TBST + 1% BSA, and incubated with AlexaFluor568 conjugated secondary antibody (1:600 in TBS + 1% BSA) for 1.5 h. Finally, coverslips were incubated with DAPI (1:1000) for 5 min to stain nuclei, washed in TBST, and mounted in N-propyl-galleate mounting medium (2% w/v in PBS/glycerol). Cells were visualized using the ×60/1.35 NA objective of an Olympus Bx63 epifluorescence microscope. Image adjustments were carried out using ImageJ software. Line scans were performed using the ColorProfiler ImageJ software plugin.

**Crystallization, data collection, and structure determination**. The NHE1ct$_{Δ92}$:CN complex (and variants) was concentrated to 8–10 mg/ml in 10 mM Tris–HCl pH 7.5, 100 mM NaCl, 1 mM CaCl$_2$, 0.5 mM TCEP. Initial crystallization trials using sitting drop vapor diffusion yielded needle-like crystals in 40% (v/v) PEG 600, 100 mM CHES/sodium hydroxide pH 9.5 at RT. Optimization was performed using the initial crystal hits as seed stock for seeding. The final crystals were obtained from seeding in 40% (v/v) PEG 600, 100 mM CHES/sodium hydroxide pH 9.5, 0.2 M MgCl$_2$. Crystals were cryoprotected using a 15-s soak in mother liquor and immediately flash frozen in liquid nitrogen. Data were collected at SSRL (beamline 12–2; NHE1ct$_{Δ92}$:CN) or the University of Arizona (Bruker liquid Gallium MetalJet X-ray Diffractometer with a Photon II CPAD detector; NHE1ct$_{Δ92RARA}$:CN and NHE1ct$_{Δ92}$:CN$_{AA}$). Data were processed using either AutoXDS[42,43], or SAINT/XPREP (Bruker AXS Inc., Madison, 2004). All three structures were phased using molecular replacement (PHASER as implemented in Phenix[44], using a CN heterodimer molecule (CNA/B; PDB ID 5SVE[26]), as the search model. A solution was obtained in space group C222$_1$; clear difference electron density for the PxIxIT and LxVP motifs was visible in the initial maps. The initial models of the complex were built without NHE1 using AutoBuild, followed by iterative rounds of refinement in PHENIX and manual building using Coot[45]. The NHE1 coordinates were then modeled, followed by additional rounds of refinement and model building. The percentage of the Ramachandran favored/allowed residues for the NHE1ct$_{Δ92}$:CN, NHE1ct$_{Δ92RARA}$:CN, and NHE1ct$_{Δ92}$:CN$_{AA}$ structures are 97.7/2.1, 97.2/2.8, and 96.2/3.8, respectively. Data collection and refinement details are provided in Supplementary Table 1. Molecular figures were generated using Pymol[46]. Linker residues that connect the NHE1 LxVP and PxIxIT motifs were modeled using Coot.

**Protein stability measurements**. Protein stability measurements report on the melting temperature of the protein (in 20 mM Tris–HCl pH 7.5, 100 mM NaCl, 1 mM CaCl$_2$, and 0.5 mM TCEP) under investigation and were performed on a Tycho NT.6 (Nanotemper) using standard capillaries (10 μl) using a 30 °C/min ramp (from 35 to 95 °C) and evaluated using the Tycho NT.6 software version 1.1.5.668.

**pNPP activity assay**. The activities of freshly prepared CN and CN$_{AA}$ were measured in assay buffer (20 mM Tris–HCl pH 7.5, 100 mM NaCl, 1 mM CaCl$_2$, 1 mM MnCl$_2$, 0.5 mM TCEP) containing varying concentrations of $p$-nitrophenyl phosphate (pNPP; 0–8000 μM). Enzymes (0.5 μM) were incubated with substrate at 30 °C for 20 min. The reaction was stopped using 0.5 M sodium phosphate and the absorbance measured at 405 nm using a plate reader (BioTek). Measured absorbance from blanks that contained substrate but no protein was subtracted from all measurements. The rate of dephosphorylation of pNPP was analyzed using the molar extinction coefficient for pNPP (18,000 M$^{-1}$ cm$^{-1}$) and an optical path

length of 0.3 cm (96-well plates). $K_m$ and $V_{max}$ were determined by fitting to the Michaelis–Menten equation, $y = V_{max}*x/(K_m + x)$; $k_{cat}$ was extracted using $y = E_t*k_{cat}*x/(K_m + x)$ (Sigma Plot 13). The catalytic efficiency was obtained as $k_{cat}/K_m$. All experiments were carried out in triplicate.

**Bioinformatics**. The CNcon database was used for screening for the TxxP in other CN substrates[26]. ScanProsite was used to identify LxVP sites using [NQDESRTH]-[YTDFILV]Lx[VPL]X as definition, which is based on the experimental 3D NFATc1$_{LxVP}$:CN complex structure and experimentally confirmed LxVP motifs. Additionally, disorder prediction via IUPRED (IUPRED ≥0.4)[47] was used to ensure that identified πφLxVP SLiMs are in IDRs. ScanProsite was further used to identify TxxP motifs for proteins in the CNcon database with an LxVP site using [S/T]xxP as the search definition. [S/T]xxP sites were also filtered for IDR behavior using different IDP predictors including IUPred. To identify the distance between C-terminal LxVP and [S/T]xxP sites, a minimum cut-off of 9 residues was used, as this distance is sufficient to span the distance between the CN LxVP-binding pocket and the CN active site.

**Measurements of pH$_i$**. Measurements of pH$_i$ were performed using the ammonium pre-pulse technique[48]. Briefly, $7*10^4–10*10^4$ PS120 WT-expressing or variant NHE1-expressing cells were seeded in 24-well plates 24 h prior to the experiment, then loaded with 2′,7′-bis-(2-carboxyethyl)-5-(and-6)-carboxyfluorescein acetoxymethylester (BCECF-AM, 1.6 μM) for 30 min at 37 °C. Cells were washed in Ringer solution (in mM, 130 NaCl, 3 KCl, 20 Hepes, 1 MgCl$_2$, 0.5 CaCl$_2$, 10 NaOH pH 7.4) twice, then bathed in Ringer again and placed in a FluoStar Optima plate reader (37 °C). Emission was measured at 520 nm and excitation at 485 nm. After baseline measurement (10 min) in Ringer, acidification was induced by washout of 20 mM NH$_4$Cl after 5 min exposure. Na$^+$-free solution (in mM, 135 NMDGCl, 3 KCl, 20 Hepes, 1 MgCl$_2$, 0.5 CaCl$_2$, pH 7.4) was applied for 1.5 min then the recovery was measured in Ringer for 10 min. Calibration was performed using the high-K$^+$/nigericin technique (in mM, 140 KCl, 10 Hepes, 1 MgCl$_2$, 0.5 CaCl$_2$, 50 μM nigericin, pH 7.4) and an eight-point calibration curve. Maximal acidification did not differ statistically between cell lines, hence recovery rate was calculated as the slope of the initial linear part of the recovery curve.

**Cell surface biotinylation**. To determine the surface fraction of all NHE1 variants, cells were incubated for 30 min at 4 °C with freshly made EZ-Link Sulfo-NHS-SS-Biotin (Life Technologies, #21331) diluted in PBS, followed by washing in cold quenching buffer (0.1 M glycine in PBS). Cells were lysed in cold RIPA buffer (150 mM NaCl, 50 mM Tris–HCl pH 7.5, 0.1% SDS, 0.5% sodium deoxycholate, 1% Igepal CA630, and Complete™ mini protease inhibitor). Samples were centrifuged (15 min, 16,000 × $g$, 4 °C), and the supernatant was adjusted to equal amounts of protein (DC assay, BioRad). A fraction of the supernatant was dissolved in 4x LDS sample buffer and saved as total lysate fraction. Remaining supernatant was incubated with prewashed streptavidine beads (Sigma-Aldrich, #S1638) for 2 h at 4 °C. Samples were washed in cold RIPA buffer, dissolved in 2x LDS sample buffer and heated for 5 min at 95 °C. Beads were pelleted (4 min, 2000 × $g$, 4 °C), the supernatant saved as the pull-down fraction and samples were processed for Western blotting.

**Antibodies**. Catalog numbers and dilutions used for WB, Western blot; IF, immunofluorescence analysis; IP, immunoprecipitation; PLA, proximity ligation assay: NHE1-54 (Santa Cruz Biotechnology, #sc-136239): WB and IF: 1:100; CNA (Cell Signaling Technology, #5530, #5532): WB 1:200; XB-17 (Mark Musch [University of Chicago]): IF, PLA 1:100; p150 (BD Bioscience, BD610473): WB 1:1000; NHE1 4e9 (Millipore-Sigma, #MAB3140); CNA (Millipore-Sigma, #C1956); CNA (Millipore-Sigma, #C1956) and IgG isotype ctrl (Cell Signaling Technology, #CS2729) for IP: all 1 μg/mg total protein; HRP-conjugated secondary antibodies (Agilent-Dako #P0447 (mouse) and #P0448 (rabbit)): WB 1:4000; Alexa fluor-conjugated secondary antibodies (ThermoFisher #A10037 (mouse) and #A10042 (rabbit)): IF 1:600.

**Quantification and statistical analysis**. ITC measurements were repeated between 2 and 4 times; reported values are the average and standard deviation for the replicated measurements. Sigma Plot 12.5/13 was used for the statistical analysis of activity assays. For cell biological data, all data are shown as representative images or as mean measurements with standard error of means (SEM) error bars, and represent at least three independent experiments unless stated otherwise. A two-tailed unpaired Student's $t$-test was applied to test for statistically significant differences between two groups. When comparing more than two groups one-way analysis of variance (ANOVA) with Dunnett's multiple comparison post-test was used, except in Supplementary Fig. 2, where two-way ANOVA was applied. *, **, *** and **** denotes $p < 0.05$, $p < 0.01$, $p < 0.001$, and $p < 0.0001$, respectively.

**Reporting summary**. Further information on research design is available in the Nature Research Reporting Summary linked to this article.

## Data availability

All NMR chemical shifts have been deposited in the BioMagResBank with the accession codes BMRB 26755 and BMRB 27812. Atomic coordinates and structure factors have been deposited in the Protein Data Bank with the accession codes 6NUC, 6NUF and 6NUU. The source data underlying Figs. 2c–j, 3b, 4b–g, 5b–h and Supplementary Figs. 3a–c, 7, 8a, b and Supplementary Tables 2 and 3 are provided as a Source Data file. All other data supporting the findings of this study are available from the corresponding authors on reasonable request.

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

## Acknowledgements

We are grateful to S.A. Sjørup and Katrine Franklin Mark for excellent technical assistance, Andreas Prestel for help in purifying protein, and Marc Severin and Signe Kramer for performing co-IP experiments. This work is supported by grants from the Lundbeck Foundation and by grant R01NS091336 from the National Institute of Neurological Disorders and Stroke to W.P., grant R01GM098482 from the National Institute of General Medicine to R.P., the Danish Council for Independent Research Natural Sciences: 4181-00344 to B.B.K. and S.F.P., and the Novo Nordisk Foundation (NNF18OC0034070) to S.F.P. We thank Villumfonden for generous support for NMR equipment. This research used beamline 12.2 at the Stanford Synchrotron Radiation Lightsource. Use of the Stanford Synchrotron Radiation Lightsource, SLAC National Accelerator Laboratory is supported by the US Department of Energy, Office of Science and Office of Basic Energy Sciences under Contract No. DE-AC02-76SF00515. The SSRL Structural Molecular Biology Program is supported by the DOE Office of Biological and Environmental Research and by the National Institutes of Health, National Institute of General Medical Sciences (including P41GM103393).

## Author contributions

R.H.-A., X.W., R.P., B.B.K., S.F.P. and W.P. developed the concept. R.H.-A. and X.W. designed, optimized, and performed in vitro phosphorylation and dephosphorylation experiments using NMR spectroscopy; X.W. designed, optimized, and performed crystallization and structure determination experiments; R.H.-A., X.W. and S.R.S. performed and analyzed ITC experiments. X.W. performed CN activity and $T_M$ experiments and analysis. R.H.-A. and X.W. performed sequence alignments for the C-terminal distal tail of NHE1 and the i + 3-binding pocket of Ser/Thr phosphatases, respectively. S.R.S. performed CN bioinformatics analysis for CN substrate identification. A.H.B., L.M.S.-F., E.P. and S.F.P. designed, performed, and analyzed cellular data. R.H.-A., X.W., S.R.S., R.P., B.B.K., S.F.P. and W.P. wrote the manuscript with comments and inputs from all co-authors.

## Additional information

**Competing interests:** The authors declare no competing interests.

**Peer Review Information:** *Nature Communications* thanks Jukka Westermarck and other anonymous reviewer(s) for their contribution to the peer review of this work. Peer reviewer reports are available.

