## [Peer Review File · Nature Communications]

Reviewers' comments:

Reviewer #1 (Remarks to the Author):

This is an excellent paper. The authors have extensively characterized the scaffolding of calcineurin by the Na⁺/H⁺ exchanger, isoform 1. This has been done by a combination of NMR, X-ray crystallography, physiological studies, protein binding studies, mutagenesis and more. The paper gives new insight into scaffolding by NHE1, and also gives novel information on ways in which phosphatases are recruited at a structural level.

There are some minor concerns.

Authors should note that T779 and S785 were identified as functionally important in 2007, J Biol Chem 282: 6292 .

Page 4, line 69, direct binding to PP1 to NHE1 has been demonstrated earlier (Biochemistry 2005, 44(15)5842)

Figure 2H, how many times was this experiment repeated. It is not as convincing as expected. (though Fig. 2I looks very convincing).

Page 36, methods, line 753, the amino acids expressed and purified should be indicated at this location.

Fig. 4C, and page 14 line 249, localization seems predominantly intracellular with some small fraction showing surface localization. This should be quantified by cell surface labeling to more definitively show plasma membrane localization. As is, the figure does not show good plasma membrane localization.

In Fig. 4G the T779A mutant appears to have about double the WT recovery rate of wild type. What is the reason for this? It appears to be at odds with the idea that phosphorylation of T779 is stimulatory. You might expect it to have basal or a lower level of activity if phosphorylation is stimulatory. This data could be more accurate if targeting levels were used to normalize activity.

Larry Fliegel

Reviewer #2 (Remarks to the Author):

Serine/Threonine (S/T) phosphorylation is the major phosphorylation event in our cells and thus understanding the molecular details that define dephosphorylation of S/T sites is of general relevance. Study by Peti laboratory uses Calcineurin (CN)-mediated dephosphorylation of NHE1 as a study example to provide a very detailed mechanistic insights how S/T phosphatase both recognize and bind the substrate and how structural assembly of phosphatase-substrate pair affects selection of dephosphorylated amino acid. Although, parts of the study are somewhat repetitive and confirmatory for previously recognized recognition mechanisms, this study significantly broadens our understanding of mechanism by which S/T phosphatase, previously considered as relatively unspecific enzymes, can reach very high substrate specificity. Thus I only have few points of criticism to this high quality of work.

Introduction:

- In.67: It is not entirely true that recognition motifs for S/T phosphatases would not been known. and that there would not be bioinformatics approaches to identify substrates. An example is

recognition sequences of B56-PP2A complexes

- Figure 2: Does introduced mutations on NHE1 change their subcellular localization? This could be an alternative explanation for differential effects

- Figure 2: Why different NHE1 mutants have differential immunoprecipitation efficiency. All IP data has to be quantified and presented as mean of at least three independent experiments and the results have to be analyzed statistically.

- Fig. 2 and other figures: It is unclear how many times experiments were repeated and whether e.g. affinities presented represent results from individual experiment and what is repeatability of reported affinities. Statistical analysis is required from multiple independent experiments.

- In. 377: The results indicating that amino acid flanking the serine and threonine residues define the specificity are clearly the most impactful and interesting outcome of this research. The authors mention that the preferred substrate peptide would have different structure, but no attempts have been made to even model how the difference looks like. For helping the reader to understand this important outcome of the research, it would be preferred that the authors would provide at least schematic presentation how they think that the flanking amino acids "decide" whether a S or T residue reach the catalytic pocket of CN and how this could be generalized to other S/T phosphatases

Reviewer #3 (Remarks to the Author):

This is a beautifully written manuscript describing comprehensive work used to demonstrate that NHE1 is a calcineurin (CN) substrate and identifying the sources of specificity. Notably, the authors show that although both the LxVP and PxIxIT motifs within NHE1 are important for both CN binding and specificity, they are not sufficient to explain the specific and rapid dephosphorylation of T779. Notably there are multiple phospho-serines nearby within the NHE1 sequence that are not efficiently dephosphorylated by CN. Through careful experimentation, the authors have been able to determine that T779 is a part of a TxxP motif that binds directly to the CN active site and confers the specificity not completely provided by the LxVP and PxIxIT motifs. This work should be of great interest to many readers given the paucity of information we have on the specificity of phosphatases.

My one concern with the work is that all phosphorylations were done using ERK2. The authors note in the Introduction that NHE1 activity is modulated by a variety of kinases, including ERK 1 and 2, p38 and JNK. Does CN only dephosphorylate residues phosphorylated by ERK2? If so, that should be stated clearly and supported by citations. Additionally, shouldn't the TxxP motif more correctly be a TPxP motif? If CN dephosphorylates residues acted upon by other kinases, then in reality the authors have identified T779 as an ERK2-dependent substrate for CN. C-terminal to T779 is S783 which is part of an SxxP motif. Is S783 in NHE1 phosphorylated by some other kinase and if so, is it a CN substrate?

Reviewer #4 (Remarks to the Author):

This work investigates the mechanism of dephosphorylation of the intrinsically disordered region of NHE1 by calcineurin. NMR spectroscopy is combined with X-ray crystallography, site-directed mutagenesis and cell experiments to demonstrate that calcineurin selectively dephosphorylates T779 in the IDP region of NHE1. It is further shown that a proline residue at position i+3 with

respect to T779 critically contributes to the selective dephosphorylation of T779. The data are of very high quality, the manuscript is beautifully illustrated and well written.

At the same time, when I finished reading the manuscript, I was wondering what actually the novel findings of these studies are and if they justify publication in a high profile journal such as Nat Comm. In my reading, the authors found that calcineurin dephosphorylates selectively T779, because it is the only threonine that was phosphorylated (other residues were serine). As the authors point out it is very well known that calcineurin preferentially dephosphorylates threonine. In addition, it is expected that residues in direct vicinity to the phosphate group influence the dephosphorylation reaction (this is well known for phosphorylation by kinases and probably not too surprising/novel for dephosphorylation).

In the discussion, the authors also state that they newly discovered a LxVP motif in the IDP region of NHE1 by sequence alignment. Again, is this really surprising? Apparently there have already been crystal structures determined of calcineurin in complex with other LxVP motifs (reference #20).

Another critical issue is the strong emphasis on the "complexity" of the dephosphorylation reaction and the contribution of the flexible 27-residue linker between the two binding motifs. I think this is strongly overemphasised and not supported by the data. First, a flexible linker connecting two binding motifs is generally expected to modulate the binding process and thus the enzymatic activity, simply because of entropic considerations. This does not require any specific binding or bound conformation (as suggested by the modelling in Fig. 3a). Second, the authors try to establish the contribution of the linker to binding and dephosphorylation by introducing site specific mutations (or competition with peptides). However, replacement of two arginine residues in the linker by two alanine residues replaces two positively charged residues by two (at least partially) hydrophobic residues, which of course will change the properties of the ligand and thus modulate transient interaction of the linker with calcineurin. Third, it is stated that this mutation results in unchanged affinity but different T_{deltaS} . In contrast, to this statement Table 1 lists different KD values when comparing wild-type and mutant NHE1, but T_{deltaS} values that are identical within the estimated errors. Thus, I do not see sufficient evidence for the importance of the linker and the complex nature of the dephosphorylation reaction. In the end, the studies boil down to the well known dephosphorylation of a threonine in an IDP region, which is influenced by residues in the direct vicinity of the phosphate group.

We would like to thank the four reviewers for their insightful comments. We appreciate the opportunity to address the questions and comments raised about our manuscript by the reviewers.

In the response to the comments/questions, we have done the following to make our responses easy to read and identify:

- included all the Reviewer comments in ***bold italics***.
- Our responses are listed immediately below each point in standard font.

Reviewer #1 (Remarks to the Author):

This is an excellent paper. The authors have extensively characterized the scaffolding of calcineurin by the Na⁺/H⁺ exchanger, isoform 1. This has been done by a combination of NMR, X-ray crystallography, physiological studies, protein binding studies, mutagenesis and more. The paper gives new insight into scaffolding by NHE1, and also gives novel information on ways in which phosphatases are recruited at a structural level.

We appreciate the strong support of our work from Dr. Fliegel.

There are some minor concerns.

Authors should note that T779 were S785 were identified as functionally important in 2007, J Biol Chem 282:6292.

We appreciated this correction and have added this citation (ref 32).

Page 4, line 69, direct binding to PP1 to NHE1 has been demonstrated earlier (Biochemistry 2005, 44(15)5842)

We appreciated this correction and have added this citation at this position in the manuscript (ref 13)

Figure 2H, how many times was this experiment repeated. It is not as convincing as expected. (though Fig. 2I looks very convincing).

The co-IP experiments were repeated 3 times. However, these experiments were done under low salt conditions and in the absence of Ca²⁺. In the process of this revision, we realized that to reflect cellular conditions, this experiment must be performed using physiological salt conditions and should assess whether an increase in the free cellular Ca²⁺ concentration [Ca²⁺]_i impacts the NHE1-calcineurin interaction. Thus, we repeated the experiment using 140 mM salt, and evaluated the interaction under resting [Ca²⁺]_i and after a [Ca²⁺]_i increase induced by ionomycin before lysis, plus 100 μM Ca²⁺ in the lysis buffer.

Under these physiologically correct conditions, a loss of interaction (albeit not statistically significant) is only seen after mutating both the LxVP and the PxlxlT motifs, with little effect observed in response to a change in $[Ca^{2+}]_i$. The new CN:NHE1 IP experiment was repeated 4 times and quantified. A representative blot is shown as Figure 2H, and quantification is shown in Figure S2 and the data are discussed throughout the manuscript.

Page 36, methods, line 753, the amino acids expressed and purified should be indicated at this location.

We agree with the reviewer that this addition makes the methods even more clear; as requested all constructs are now fully described in the methods.

Fig. 4C, and page 14 line 249, localization seems predominantly intracellular with some small fraction showing surface localization. This should be quantified by cell surface labeling to more definitively show plasma membrane localization. As is, the figure does not show good plasma membrane localization.

We agree with the reviewer and thank him for the opportunity to improve the figure. We now provide even clearer images in Figure 4C and have also added line scans below each image to illustrate membrane localization. Furthermore, we quantified membrane localization by measuring cell surface biotinylation. This data is included as a new Figure S7. Collectively, these data show that membrane localization of the NHE1 variants is identical to that of WT NHE1.

In Fig. 4G the T779A mutant appears to have about double the WT recovery rate of wild type. What is the reason for this? It appears to be at odds with the idea that phosphorylation of T779 is stimulatory. You might expect it to have basal or a lower level of activity if phosphorylation is stimulatory. This data could be more accurate if targeting levels were used to normalize activity.

We show now in detail (see response to the comment above) that there is no detectable difference in membrane localization between WT NHE1 and T779A NHE1, hence, the differences are unlikely to reflect variations in surface expression. The mutation does not have any *in silico* measured effect on helicity (e.g. by using Agadir) so the differences are likely not due to any local structural change. Thus, we currently do not have any firm explanation for this effect, but it very likely reflects the balance between the phosphorylated and the non-phosphorylated states that controls the activity of NHE1.

Larry Fliegel

Reviewer #2 (Remarks to the Author):

Serine/Threonine (S/T) phosphorylation is the major phosphorylation event in our cells and thus understanding the molecular details that define dephosphorylation of S/T sites is of general relevance. Study by Peti laboratory uses Calcineurin (CN)-mediated dephosphorylation of NHE1 as a study example to provide a very detailed mechanistic insights how S/T phosphatase both recognize and bind the substrate and how structural assembly of phosphatase-substrate pair affects selection of dephosphorylated amino acid. Although, parts of the study are somewhat repetitive and confirmatory for previously recognized recognition mechanisms, this study significantly broadens our understanding

of mechanism by which S/T phosphatase, previously considered as relatively unspecific enzymes, can reach very high substrate specificity. Thus I only have few points of criticism to this high quality of work.

We appreciate the strong support of our work by the reviewer.

Introduction:

- In.67: It is not entirely true that recognition motifs for S/T phosphatases would not been known. and that there would not be bioinformatics approaches to identify substrates. An example is recognition sequences of B56-PP2A complexes

We appreciate this comment. We referred to the fact that no defined substrate recognition motifs for S/T phosphatases for their active/catalytic sites have been reported. It is true that substrate binding sites away from the phosphatase active site have been reported and used for bioinformatics analysis for substrates. Indeed, our group has done this for CN (LxVP motif) and the B565-PP2A LxxIxE motif. Furthermore other groups have used the PxIxIT site etc. We have now clarified this in the next (i.e., page 5, “we define the key interactions that generate this CN selectivity” to “we define the key interactions that generate this CN selectivity at the CN active site”).

- Figure 2: Does introduced mutations on NHE1 change their subcellular localization? This could be an alternative explanation for differential effects

We agree this could be an alternative explanation. To quantify the membrane localization of the NHE1 variants, we now provide improved images and furthermore have added line scans to show membrane localization (in Figure 4C). We also quantified membrane localization from measurements of cell surface biotinylation and added this as a new Figure S7. These data show that membrane localization of the variants is not reduced compared to wild type.

- Figure 2: Why different NHE1 mutants have differential immunoprecipitation efficiency. All IP data has to quantified and presented as mean of at least three independent experiments and the results have to be analyzed statistically.

We agree that the pulldown of NHE1 variants with CN shows higher consistency than that of CN with NHE1, likely reflecting differences in antibody affinity and/or that NHE1 is overexpressed, while CN is endogenous. We thank you for suggesting quantification of the immunoprecipitation data. In the process of this revision, we realized that to reflect cellular conditions, this experiment must be performed under physiological salt conditions and should also assess whether an increase in the free cellular Ca^{2+} concentration $[Ca^{2+}]_i$ impacts the NHE1-calcineurin interaction. Thus, we repeated the experiment using 140 mM salt and evaluated the interaction under resting $[Ca^{2+}]_i$ and after a $[Ca^{2+}]_i$ increase induced by ionomycin before lysis, plus 100 μM Ca^{2+} in the lysis buffer. Under these physiologically conditions, a loss of interaction (albeit not statistically significant) is only seen after mutating both the LxVP and the PxIxIT motifs, with little effect observed in response to a change in $[Ca^{2+}]_i$. The new CN-NHE1 IP experiment was repeated 4 times and quantified. A representative blot is shown as Figure 2H, and quantification is shown in Figure S2.

- Fig. 2 and other figures: It is unclear how many times experiments were repeated and whether e.g. affinities presented represent results from individual experiment and what is

repeatability of reported affinities. Statistical analysis is required from multiple independent experiments.

Both experiments shown in the original Figures 2H and 2I were repeated in triplicate. As noted above, we have now performed physiologically correct co-IP experiments. These experiments have been repeated 4 times, a blot of mean values with S.E.M. error bars are shown in Figure S2, and data have been statistically analyzed, as described in the new figure legend.

- In. 377: The results indicating that amino acid flanking the serine and threonine residues define the specificity are clearly the most impactful and interesting outcome of this research. The authors mention that the preferred substrate peptide would have different structure, but no attempts have been made to even model how the difference looks like. For helping the reader to understand this important outcome of the research, it would be preferred that the authors would provide at least schematic presentation how they think that the flanking amino acids “decide” whether a S or T residue reach the catalytic pocket of CN and how this could be generalized to other S/T phosphatases

We thank the reviewer that we have not made this point clear enough. We do not think that the preferred substrate peptide has a different structure – it has a different primary sequence/structure. *The key driver of this sequence is an i+3 Pro residue* (with i being either Ser or Thr). Indeed, we show a model of this interaction in Figures 5I and 5J, showing how the NHE1 TxxP motif most likely interacts with CN. Furthermore, we also show (Figure 5J and 5K) that CN residues H155 and Y159 form the i+3 proline-binding pocket (i.e., cradling the proline residue from the TxxP motif). However, in spite of these data, we cannot speculate on why Thr is preferred over Ser. Indeed, as our results show, the i+3 Pro is the dominant specificity determinant for the selection of the dephosphorylation residue.

Lastly, and equally importantly– and as stated in the manuscript - H155 and Y159 are *not conserved in any other PPP family member* and thus this pocket (and binding specificity determinant) unique to CN (see sequence alignment in Figure S5D). Clearly, this uniqueness allows on to leverage this discovery in order to identify additional CN substrates, which we have done and described in the discussion.

Reviewer #3 (Remarks to the Author):

This is a beautifully written manuscript describing comprehensive work used to demonstrate that NHE1 is a calcineurin (CN) substrate and identifying the sources of specificity. Notably, the authors show that although both the LxVP and PxlxlT motifs within NHE1 are important for both CN binding and specificity, they are not sufficient to explain the specific and rapid dephosphorylation of T779. Notably there are multiple phosphoserines nearby within the NHE1 sequence that are not efficiently dephosphorylated by CN. Through careful experimentation, the authors have been able to determine that T779 is a part of a TxxP motif that binds directly to the CN active site and confers the specificity not completely provided by the LxVP and PxlxlT motifs. This work should be of great interest to many readers given the paucity of information we have on the specificity of phosphatases.

We thank the reviewer for her/his substantial support for our work.

1. My one concern with the work is that all phosphorylations were done using ERK2. The

authors note in the Introduction that NHE1 activity is modulated by a variety of kinases, including ERK 1 and 2, p38 and JNK. Does CN only dephosphorylate residues phosphorylated by ERK2? If so, that should be stated clearly and supported by citations.

We appreciate the comment of the reviewer. We focused on ERK2 in this manuscript to highlight the *dephosphorylation events* by CN. Nevertheless, we have certainly tested if p38 or JNK can also phosphorylate NHE1. Indeed, all three MAPKs – ERK2, p38 α and JNK – show a similar phosphorylation pattern of NHE1 with T779 always being phosphorylated. A figure summarizing this finding has been added to the supporting information Figure S6 to alleviate this concern.

2. Additionally, shouldn't the TxxP motif more correctly be a TPxP motif? If CN dephosphorylates residues acted upon by other kinases, then in reality the authors have identified T779 as an ERK2-dependent substrate for CN.

The reviewer has a good point – for NHE1 we could call this a TPxP motif. However, additional experiments beyond the data described in this manuscript show that TxxP is a better description of the motif. Indeed, our structural model (Figure 5I-K) highlights that there is little complementary surface for the two X residues. Furthermore, other non-proline directed kinases can also phosphorylate TxxP motifs, which are recognized by CN. Indeed, varying the xx residues also changes the interaction strength (these are ongoing studies in the laboratory; namely, efforts to more precisely define the motif by identifying novel CN substrates). Thus, instead of renaming the motif in a future publication, we would prefer to refer to it as the TxxP motif as it will allow for easier cross referencing in the future and a more comprehensive understanding of the role of this motif in CN-mediated dephosphorylation.

3. C-terminal to T779 is S783 which is part of an SxxP motif. Is S783 in NHE1 phosphorylated by some other kinase and if so, is it a CN substrate?

Again we appreciate the very insightful question from the reviewer. She/he is certainly correct that there is a second SxxP motif directly C-terminal to the TxxP motif. The corundum of studying this site is that it is currently unknown if this is a physiologically relevant phosphorylation site, i.e. an unknown kinase is responsible for its phosphorylation. Predictions suggest that kinases as diverse as GSK3, CDK2 and even MAPKs (despite the fact that there is no proline C-terminal of S783 and none of the three tested MAPKs phosphorylated it) have a similar probability for phosphorylating this residue. Furthermore, no manuscripts that specifically focus on this residue have been published (see phosphositeplus; <https://www.phosphosite.org>). However, the site has been found to be phosphorylated in high-throughput phosphoproteomics studies without stating which kinase is responsible for the phosphorylation. It is for these reasons that we have not attempted to identify: (1) the kinase and (2) if S783 and the SxxP motif is a CN substrate. It must be pointed out that this is one of the most difficult aspects of understanding protein phosphatase substrates – the need for knowing kinase:phosphatase pairs instead of only a kinase.

Reviewer #4 (Remarks to the Author):

This work investigates the mechanism of dephosphorylation of the intrinsically disordered region of NHE1 by calcineurin. NMR spectroscopy is combined with X-ray crystallography, site-directed mutagenesis and cell experiments to demonstrate that calcineurin selectively dephosphorylates T779 in the IDP region of NHE1. It is further shown that a proline residue at position i+3 with respect to T779 critically contributes to the selective

dephosphorylation of T779. The data are of very high quality, the manuscript is beautifully illustrated and well written.

We appreciate the kind words for our work and the strong support of this reviewer.

At the same time, when I finished reading the manuscript, I was wondering what actually the novel findings of these studies are and if they justify publication in a high profile journal such as Nat Comm. In my reading, the authors found that calcineurin dephosphorylates selectively T779, because it is the only threonine that was phosphorylated (other residues were serine). As the authors point out it is very well known that calcineurin preferentially dephosphorylates threonine. In addition, it is expected that residues in direct vicinity to the phosphate group influence the dephosphorylation reaction (this is well known for phosphorylation by kinases and probably not too surprising/novel for dephosphorylation).

Indeed, what we show here is that threonine selectivity is not the significant determinant for selectivity. Indeed when we replace T779 with S779 *it is still the residue that is most rapidly dephosphorylated* (Figure 4B and 5B). To emphasize this, the header of the paragraph that describes the findings is: “The specificity of CN for pT779 is not explained by Thr preference”.

We agree it would be invaluable to the scientific community to have well-defined, precise recognition sequences for phosphatases, similar to those identified for kinases. Unfortunately this is not the case. Indeed, many carefully performed studies (e.g. Li, X, Wilmanns, M, Thornton, J & Köhn M (2013) Elucidating human phosphatase-substrate networks, Sci Signal., 14;6(275):rs10) have shown that the family of PPPs do not have well-defined PPP-specific substrate recognition sequence. Thus, the identification of the CN-specific TxxP sequence is truly novel and we are certain will be used in the future, together with the PxlIT and LxVP SLiM sequence, to understand the full CN proteome.

1. In the discussion, the authors also state that they newly discovered a LxVP motif in the IDP region of NHE1 by sequence alignment. Again, is this really surprising? Apparently there have already been crystal structures determined of calcineurin in complex with other LxVP motifs (reference #20).

The reviewer is correct – we identified a previously unidentified LxVP motif in NHE1. Next, we also biochemically and structurally confirmed it and showed its functional importance in cellular experiments. It is true that there is a structure of CN with an LxVP substrate peptide (from NFAT1c) and a structure of CN in complex with a viral inhibitory protein A238L. Our structure shows – surprisingly – that the i-3 residue (i being the L of the LxVP motif) contributes to the interaction with CN. This expands our understanding of this important interaction.

We understand that it can seem repetitive to have multiple structures, but indeed comparisons allow for critical biological insights to be gained. While we have a large number of kinase, ribosome or now even GCPR structures, each structure – when carefully analyzed – is a building block towards a comprehensive understanding of the underlying biology that is driven by protein:protein interactions.

2. Another critical issue is the strong emphasis on the "complexity" of the dephosphorylation reaction and the contribution of the flexible 27-residue linker between the two binding motifs. I think this is strongly overemphasised and not supported by the data. First, a flexible linker connecting two binding motifs is generally expected to modulate the binding process and thus the enzymatic activity, simply because of entropic

considerations. This does not require any specific binding or bound conformation (as suggested by the modelling in Fig. 3a).

We are not aware of literature that shows that a linker connecting SLiM motifs leads to a modulation of substrate binding. We agree that the linker might not adopt any of the modeled conformations, but we felt that a model helps to understand this complex mechanism to be easier understood and would prefer to keep it.

Second, the authors try to establish the contribution of the linker to binding and dephosphorylation by introducing site specific mutations (or competition with peptides). However, replacement of two arginine residues in the linker by two alanine residues replaces two positively charged residues by two (at least partially) hydrophobic residues, which of course will change the properties of the ligand and thus modulate transient interaction of the linker with calcineurin.

The reviewer is correct and it is exactly the point that we highlight – the linker is ‘activated’ by the 2 arginine residues that allow it to ‘glide’, via electrostatic interactions over the CN surface. Charge:charge interactions are emerging as a critical new mode of biomolecular interactions, especially for understanding the interaction of IDPs with folded binding partners (like CN) and even IDPs. This is a powerful example that shows how these types interactions influence biological functions.

Third, it is stated that this mutation results in unchanged affinity but different TdeltaS, In contrast, to this statement Table 1 lists different KD values when comparing wild-type and mutant NHE1, but TdeltaS values that are identical within the estimated errors. Thus, I do not see sufficient evidence for the importance of the linker and the complex nature of the dephosphorylation reaction.

We appreciate the critical review of our work – but Table 1 shows clearly different TdeltaS values that are statistically different and with opposite sign.

	K_D (nM)	ΔH (kcal* mol^{-1})	$T\Delta S$ (kcal* mol^{-1})
NHE1 $ct_{\Delta 92}$ (NHE1ct I680-S723)	21 ± 2	-8.5 ± 0.4	2.0 ± 0.3
Charge neutral NHE1ct dynamic linker			
NHE1 $ct_{\Delta 92RARA}$	21 ± 2	-12.2 ± 0.1	-1.7 ± 1.2

3. In the end, the studies boil down to the well known dephosphorylation of a threonine in an IDP region, which is influenced by residues in the direct vicinity of the phosphate group.

We appreciate the input of the reviewer – and while the summary is certainly correct – it is very generalized.

All requested changes have been performed and the manuscript has been significantly revised.

REVIEWERS' COMMENTS:

Reviewer #1 (Remarks to the Author):

The authors have answered all my queries appropriately and I have no other hesitation in recommending the manuscript for publication.

Reviewer #2 (Remarks to the Author):

The authors have done all the requested additional experiments and provide adequate responses to all my questions. Thus I do not have any further criticism towards the work.

Reviewer #3 (Remarks to the Author):

The authors have done an excellent job of addressing my concerns. I have no further concerns.